# Score Based Error Correcting Code Decoder

**Alon Helvits** [1]  **Eliya Nachmani** [1]

## Abstract

Error-correcting codes enable reliable communication, yet practical soft decoding remains challenging across code families and block lengths. We propose SB-ECC, a score-based decoder that casts decoding as continuous-time denoising. A neural denoiser defines a probability-flow ordinary differential equation (ODE) that iteratively updates the noisy channel observation toward a valid codeword, guided by parity constraints. The model is trained across noise levels without time/SNR conditioning, enabling inference without SNR estimation and supporting a direct latency–accuracy trade-off controlled by the ODE solver budget. We use the raw signed channel observation as input for learning a continuous denoising field. Across $42$ code/SNR settings, SB-ECC achieves the best BER in $39/42$ entries, with an average SNR gain of $0.17\,\mathrm{dB}$ and a maximum gain of $0.46\,\mathrm{dB}$ over the strongest competing baseline, we showed that swapping the solver from Euler to DPM preserves $-\ln(\mathrm{BER})$ while reducing end-to-end decoding time by $8.86\%$ on average (up to $12.82\%$).

## 1. Introduction

Reliable digital communication relies on Error-Correcting Codes (ECC) that allow a receiver to reconstruct a transmitted codeword from a noisy channel observation. Although Maximum-Likelihood (ML) decoding is optimal, it is NP-hard for general linear block codes (Berlekamp et al., 1978), making practical soft decoding a central challenge for many code families and block lengths. Classical iterative decoders, most notably belief propagation (BP) (Richardson & Urbanke, 2002), perform approximate marginalization via message passing on a code's factor/Tanner graph (Gallager,

1962; Kschischang et al., 2002).

Deep learning (LeCun et al., 2015) emerged as a compelling alternative for decoding, with two dominant lines of work. *Model-based* neural decoders start from an iterative algorithm (e.g., BP or min-sum) and "unroll" a fixed number of iterations into a trainable network, often learning iteration-dependent weights or offsets while preserving Tanner-graph structure (Nachmani et al., 2016; Cammerer et al., 2017; Lugosch & Gross, 2017; Nachmani et al., 2018; Lian et al., 2019; Buchberger et al., 2021). These methods benefit from strong inductive bias, but their computation and connectivity remain tied to the underlying iterative procedure, which can limit expressivity and scalability.

*Model-free* decoders instead learn the decoding function directly from data using different neural network architectures. Early model-free neural decoders therefore focused on short block lengths, learning a direct mapping from raw channel observations $\mathbf{y}$ (or log-likelihood ratios (LLRs)) to the transmitted bits (Seo et al., 2018b; Leung et al., 2019b; Matsumine & Ochiai, 2024). A major difficulty is the combinatorial explosion of the output space. For a generic $(n, k)$ code, the code-book size $|\mathcal{C}| = 2^k$ scales exponentially with the information length. Consequently, the available training data is inevitably sparse relative to the total volume of the codeword space, often leading to overfitting when training on raw channel observations(Bennatan et al., 2018). A key practical step that enabled modern model-free decoders is a symmetry-preserving pre-processing pipeline that utilizes reliability features, such as the magnitude $|\mathbf{y}|$ of the received signal $\mathbf{y}$ together with parity information (e.g. hard-syndrome, soft-syndrome (Lau et al., 2025) ), allowing training on a canonical transmitted codeword under BPSK symmetry (Bennatan et al., 2018). This paradigm enabled Transformer-based decoders that incorporate code structure through attention masking guided by the parity-check matrix. Error Correction Code Transformer (ECCT) introduced masked self-attention constrained by the parity-check graph to model long-range dependencies (Choukroun & Wolf, 2022b; Levy et al., 2024), and CrossMPT (Park et al., 2024) further improves scalability by treating magnitude and parity information as distinct modalities coupled via masked cross-attention blocks that emulate variable/check message exchange.

[1]School of Electrical and Computer Engineering (ECE), Ben Gurion University, Beer Sheva, Israel. Correspondence to: Alon Helvits <alonhel@post.bgu.ac.il>, Eliya Nachmani <eliyanac@bgu.ac.il>.

*Proceedings of the $43^{rd}$ International Conference on Machine Learning*, Seoul, South Korea. PMLR 306, 2026. Copyright 2026 by the author(s).

In parallel, diffusion and score-based generative models have achieved remarkable success by learning to reverse a gradual noising process (Sohl-Dickstein et al., 2015; Song & Ermon, 2019; Ho et al., 2020; Song et al., 2020b). While these foundational works were developed primarily for continuous data such as images, diffusion models have since been successfully adapted to other modalities, including audio generation (Kong et al., 2020) and text generation (Li et al., 2022). Denoising Diffusion Error Correction Codes (DDECC) (Choukroun & Wolf, 2022a) adapted this viewpoint to channel decoding by interpreting transmission over noise as a forward diffusion process and decoding as an iterative denoising trajectory guided by code constraints, while still relying on magnitude-based reliability preprocessing.

In this work, we revisit the standard preprocessing choice from the perspective of score-based decoding. Mapping $\mathbf{y} \mapsto |\mathbf{y}|$ folds the observation space and removes directional information that may be beneficial when learning a continuous guidance field in $\mathbb{R}^n$. Conceptually, this also changes the learning problem, while many model-free decoders are trained to directly predict bit logits/LLRs (Choukroun & Wolf, 2022a;b; Park et al., 2024), we train a continuous denoising model that regresses an additive-noise (equivalently, score) vector field in $\mathbb{R}^n$ (Song et al., 2020b). We therefore propose a score-based decoding framework, **SB-ECC** (*Score-Based Error-Correcting Code Decoder*), that trains and operates directly on the *signed* received signal $\mathbf{y}$. We formulate decoding under a variance-exploding (VE) construction in continuous time (Song et al., 2020b), and train a neural score model to predict additive noise (equivalently, the score under the VE marginal) conditioned on the noisy observation and code constraints.

A practical challenge in this setting is that the effective noise level (and thus the corresponding diffusion "time") is not known at inference time, since the channel SNR is typically unavailable. Rather than estimating time explicitly, we use a *time-unconditional* score model trained across a range of noise levels, together with a simple *linear noise schedule*. This removes any need for SNR side information and avoids committing to a specific test-time noise calibration. Our approach naturally supports flexible inference, the number of solver steps (or function evaluations) can be chosen to trade accuracy for latency without changing the trained model, and without requiring a time/SNR input.

Our main contributions are:

1. We introduce SB-ECC, a time-unconditional score-based decoder for linear block codes that learns a continuous additive-noise (score) field directly on the signed channel observation and performs decoding via Probability-Flow ODE (PF-ODE) integration guided by parity constraints.

2. We speed up PF-ODE decoding by replacing the Euler solver with DPM-Solver while preserving performance. This is enabled by our linear noise schedule and uniform $\sigma$-space discretization, which are naturally suited to time-unconditional decoding.

3. Across multiple code families and SNRs, we improve over strong neural baselines, achieving the best BER in $39/42$ entries with an average SNR gain of $0.17\,\mathrm{dB}$ and a maximum gain of $0.46\,\mathrm{dB}$ over the strongest competing baseline.

**Reproducibility.** Code for training and evaluation is available at `https://github.com/alonhelvits/SB-ECC`.

## 2. Related Works

Neural decoders for error-correcting codes are often divided into *model-based* (model-driven) and *model-free* approaches. Model-based methods start from iterative decoders such as belief propagation (BP) or min-sum (MS) and unroll a fixed number of message-passing iterations into a trainable network while preserving the Tanner-graph computation (Nachmani et al., 2016; Cammerer et al., 2017; Lugosch & Gross, 2017; Nachmani et al., 2018; Lian et al., 2019; Buchberger et al., 2021). For BP, (Nachmani et al., 2016) learns edge- and/or iteration-dependent parameters within an unrolled Tanner-graph decoder. For MS, offset min-sum variants retain the same message-passing structure but learn additive *offset* corrections (Lugosch & Gross, 2017). More expressive hybrids modify the update rules beyond simple scalings/offsets, e.g., via hypernetwork-style parameterizations, autoregressive messaging, or learned decimation mechanisms (Nachmani & Wolf, 2019; 2021; Buchberger et al., 2021).

Unlike model-based decoders, *model-free* neural decoders learn the mapping from the noisy channel observation directly using flexible architectures, without committing to a particular iterative algorithm. This line of work dates back to early neural decoding efforts (e.g., NN realizations/approximations of classical decoders and recurrent decoders for convolutional codes) (Wang & Wicker, 1996; Hamalainen & Henriksson, 1999). Modern deep-learning-era model-free decoders typically use fully connected or recurrent architectures that ingest the received soft vector $\mathbf{y}$ (O'shea & Hoydis, 2017; Gruber et al., 2017; Seo et al., 2018a). While these approaches can perform well on short block lengths, they often face sample-complexity and generalization challenges as $n$ grows, and their performance can be sensitive to architecture choice (MLP vs CNN vs RNN/LSTM) and latency constraints (Lyu et al., 2018; Sattiraju et al., 2018; Leung et al., 2019a).

The introduction of magnitude (reliability) and syndrome

preprocessing often framed as learning a *multiplicative* noise model was crucial for mitigating overfitting and enabling model-free decoders to scale beyond very short codes (Ben-natan et al., 2018; Kamassury & Silva, 2020; Artemasov et al., 2023). Complementary directions for improving robustness and generalization include fast adaptation via meta-learning and transfer learning (Jiang et al., 2019; Lee et al., 2020). Building on this paradigm, Choukroun & Wolf (2022b) introduced the Error Correction Code Transformer (ECCT), where masked self-attention injects parity-check structure into a Transformer decoder, marking a major step in demonstrating that model-free neural decoders can outperform classical and model-based baselines. Subsequent work strengthened the masking/conditioning mechanism with systematic and double-masking strategies (Park et al., 2023), explored joint encoder–decoder (end-to-end code + decoder) optimization (Choukroun & Wolf, 2024b), proposed a foundation model aimed at stronger zero-shot generalization across codes and lengths (Choukroun & Wolf, 2024a), and introduced cross-attention/message-passing variants that improve performance and efficiency on longer codes (Park et al., 2024; Cohen et al., 2025). In parallel, generative diffusion-based decoders such as Denoising Diffusion Error Correction Codes (DDECC) formulate decoding as iterative denoising with code constraints, offering a complementary paradigm with strong results across multiple code families (Choukroun & Wolf, 2022a). More recently, Lei et al. (2025) proposed an error-correction consistency-flow model that distills the denoising trajectory into one-step decoding, emphasizing low-latency inference as an alternative to iterative diffusion-based decoding.

Diffusion and score-based generative models evolved from early formulations that learn a reverse-time denoising process for a progressive noising (diffusion) forward process (Sohl-Dickstein et al., 2015), to practical large-scale diffusion models such as DDPM (Ho et al., 2020) and a series of improved training/sampling variants (Song et al., 2020a; Nichol & Dhariwal, 2021; Dhariwal & Nichol, 2021; Rombach et al., 2022; Saharia et al., 2022). In parallel, score-based generative modeling framed generation as estimating $\nabla_{\mathbf{x}} \log p_t(\mathbf{x})$ across noise levels (Hyvärinen, 2005; Vincent, 2011; Song & Ermon, 2019), and later unified in continuous time through the Score-SDE framework, where samples are obtained by numerically solving a reverse-time Stochastic Differential Equation (SDE) or its probability-flow Ordinary Differential Equation (ODE) (Song et al., 2020b). A major practical driver has been sampler/solver design, where the number of function evaluations trades off quality and latency. Examples include standard discretizations used in score-based sampling (e.g., Euler–Maruyama / predictor–corrector) (Song et al., 2020b) and EDM-style design choices such as Heun-based samplers and reparameterizations/preconditioning (Karras et al., 2022), and dedicated

high-order ODE solvers such as PNDM and DPM-Solver (Liu et al., 2022; Lu et al., 2022), as well as subsequent refinements (e.g., DPM-Solver++) (Zheng et al., 2023). While these methods were first developed and popularized primarily for high-fidelity image generation (Ho et al., 2020; Song et al., 2020a; Nichol & Dhariwal, 2021; Dhariwal & Nichol, 2021; Rombach et al., 2022; Saharia et al., 2022), diffusion/score-based modeling has since been successfully adapted to other modalities, including audio and speech synthesis (Chen et al., 2020; Kong et al., 2020; Popov et al., 2021), language and other discrete data (Austin et al., 2021; Li et al., 2022), and video generation (Ho et al., 2022a;b; Blattmann et al., 2023a;b).

## 3. Preliminaries and Background

Here we provide necessary background on error correction coding, diffusion and score-based models.

### 3.1. Error Correction Codes

We consider a standard coded transmission using a binary linear block code $C$. The code is specified by a generator matrix $G \in \{0,1\}^{k \times n}$ and a parity-check matrix $H \in \{0,1\}^{(n-k) \times n}$ such that $GH^{\top} = 0$ over GF(2) (equivalently, $H\mathbf{x}^{\top} = \mathbf{0}$ for all $\mathbf{x} \in C$) (Richardson & Urbanke, 2008). Given a message $\mathbf{m} \in \{0,1\}^k$, the encoder produces a codeword $\mathbf{x} = \mathbf{m}G \in C \subseteq \{0,1\}^n$ satisfying $H\mathbf{x}^{\top} = \mathbf{0}$.

We focus on Binary Phase Shift Keying (BPSK) modulation over Additive White Gaussian Noise (AWGN), as commonly adopted in deep decoders (Choukroun & Wolf, 2022b). Let $\mathbf{x}_s \in \{\pm 1\}^n$ denote the BPSK-modulated codeword (e.g., $\mathbf{x}_s = \mathbf{1} - 2\mathbf{x}$). The receiver observes

$$\mathbf{y} = \mathbf{x}_s + \mathbf{z}, \tag{1}$$

where $\mathbf{z} \sim \mathcal{N}(\mathbf{0}, \sigma_{\text{ch}}^2 I_n)$ and $\sigma_{\text{ch}} > 0$ is the channel noise standard deviation determined by the channel condition, yielding a soft observation $\mathbf{y} \in \mathbb{R}^n$.

A convenient baseline is the hard-decision estimate obtained by element-wise thresholding: $\hat{\mathbf{x}}_s = \text{sign}(\mathbf{y}) \in \{\pm 1\}^n$, mapped back to bits $\mathbf{y}_b = \text{bin}(\hat{\mathbf{x}}_s) \in \{0,1\}^n$. Here, $\text{sign}(y_i) = +1$ if $y_i \geq 0$ and $-1$ otherwise, and $\text{bin}(+1) = 0$ and $\text{bin}(-1) = 1$ (applied element-wise). Parity consistency of this hard decision is tested via the syndrome $\mathbf{s}(\mathbf{y}) \triangleq H\mathbf{y}_b^{\top} \in \{0,1\}^{n-k}$ over GF(2). If $\mathbf{s}(\mathbf{y}) = \mathbf{0}$ then $\mathbf{y}_b \in C$ is a valid codeword; otherwise, $\mathbf{s}(\mathbf{y}) \neq \mathbf{0}$ indicates violated constraints. When the parity checks fail, a decoder $f : \mathbb{R}^n \rightarrow \{0,1\}^n$ uses the soft observation $\mathbf{y}$ to produce a refined estimate of the transmitted codeword $\hat{\mathbf{x}} = f(\mathbf{y})$. The decoder output is validated by the same parity checks via its syndrome $H\hat{\mathbf{x}}^{\top}$, and successful decoding corresponds to $H\hat{\mathbf{x}}^{\top} = \mathbf{0}$.

**Algorithm 1** Training procedure for Score-Based Error Correction Codes

**Require:** Batch $\mathbf{x}_0$, schedule $\sigma(\cdot)$, learning rate $\eta$
1: Initialize parameters $\theta$
2: **repeat**
3:     Sample $t \sim \mathcal{U}(0,1)$ and $\boldsymbol{\epsilon} \sim \mathcal{N}(\mathbf{0}, \mathbf{I})$
4:     $\sigma \leftarrow \sigma(t), \quad \mathbf{y} \leftarrow \mathbf{x}_0 + \sigma\,\boldsymbol{\epsilon}$
5:     $\mathbf{s} \leftarrow \mathbf{H}\,\mathrm{bin}(\mathrm{sign}(\mathbf{y}))^\top \bmod 2$
6:     $\mathcal{L}_\epsilon \leftarrow \|\hat{\boldsymbol{\epsilon}}_\theta(\mathbf{y}, \mathbf{s}) - \boldsymbol{\epsilon}\|_2^2$
7:     $\theta \leftarrow \theta - \eta\,\nabla_\theta \mathcal{L}_\epsilon$
8: **until** convergence

**Algorithm 2** Decoding with early exit

**Require:** Received vector $\mathbf{y} \in \mathbb{R}^n$, parity-check matrix $\mathbf{H} \in \{0,1\}^{(n-k)\times n}$, steps $N_{\text{steps}}$, endpoints $\sigma_{\max} \to \sigma_{\min}$
1: $\Delta\sigma \leftarrow (\sigma_{\max} - \sigma_{\min})/N_{\text{steps}}$
2: $\mathbf{x}^{(0)} \leftarrow \mathbf{y}$
3: **for** $i = 0, \dots, N_{\text{steps}} - 1$ **do**
4:     $\hat{\mathbf{c}}_s \leftarrow \mathrm{sign}(\mathbf{x}^{(i)})\ \{\hat{\mathbf{c}}_s \in \{\pm 1\}^n\}$
5:     $\hat{\mathbf{c}} \leftarrow \mathrm{bin}(\hat{\mathbf{c}}_s)\ \{\hat{\mathbf{c}} \in \{0,1\}^n\}$
6:     $\mathbf{s} \leftarrow \mathbf{H}\hat{\mathbf{c}}^\top \bmod 2\ \{\mathbf{s} \in \{0,1\}^{n-k}\}$
7:     **if** $\mathbf{s} = 0$ **then**
8:       **break**
9:     **end if**
10:     $\hat{\epsilon} \leftarrow \hat{\boldsymbol{\epsilon}}_\theta(\mathbf{x}^{(i)}, \mathbf{s})$
11:     $\mathbf{x}^{(i+1)} \leftarrow \mathbf{x}^{(i)} - \Delta\sigma\,\hat{\epsilon}$
12: **end for**
13: **return** $\mathrm{bin}(\mathrm{sign}(\mathbf{x}^{(i)}))$

### 3.2. Diffusion and Score-Based Generative Models

**Diffusion Models (Discrete-Time View).** Diffusion models specify a *forward* noising process that forms a Markov chain $q(\mathbf{x}_{1:T} \mid \mathbf{x}_0) = \prod_{t=1}^{T} q(\mathbf{x}_t \mid \mathbf{x}_{t-1})$, where each transition gradually corrupts the signal by adding Gaussian noise (Ho et al., 2020). Because these Gaussian transitions compose in closed form, one can equivalently write the marginal perturbation at (continuous) time $t$ as

$$\mathbf{x}_t = \alpha(t)\,\mathbf{x}_0 + \sigma(t)\,\varepsilon, \qquad \varepsilon \sim \mathcal{N}(\mathbf{0}, I_n), \qquad (2)$$

where $\alpha(t)$ and $\sigma(t)$ define the noise schedule. Many diffusion models are trained to predict the injected noise $\varepsilon$ (or equivalently $\mathbf{x}_0$), which is closely related to the score under Gaussian perturbations (Song et al., 2020b; Ho et al., 2020; Karras et al., 2022). In this work we use the continuous-time score-based formulation below, which expresses the same Gaussian perturbation family via a stochastic differential equation (SDE) and enables sampling through its associated probability-flow ODE (Song et al., 2020b).

**Forward Diffusion as an SDE.** Score-based generative models define a continuous noising process over $t \in [0, T]$ via an Itô SDE (Song et al., 2020b):

$$d\mathbf{x}_t = f(\mathbf{x}_t, t)\,dt + g(t)\,d\mathbf{w}_t, \qquad (3)$$

where $\mathbf{x}_t \in \mathbb{R}^n$ is the state at time $t$, $f(\mathbf{x}_t, t)$ is the drift field, $g(t)$ is the diffusion amplitude, and $\mathbf{w}_t$ is a standard $n$-dimensional Wiener process. The terminal time $T$ controls the overall noise level. A common choice is the variance-exploding (VE) SDE (Song et al., 2020b), where $f(\mathbf{x}, t) = \mathbf{0}$ and the noise scale $\sigma(t)$ increases with $t$. Choosing $g(t)$ such that $g(t)^2 = \frac{d\sigma(t)^2}{dt}$ yields the marginal perturbation form

$$\mathbf{x}_t = \mathbf{x}_0 + \sigma(t)\varepsilon, \qquad \varepsilon \sim \mathcal{N}(\mathbf{0}, I_n). \qquad (4)$$

This is particularly natural for channel decoding, the AWGN channel in (1) corresponds to a VE perturbation at a specific noise level, i.e., there exists $t^\star$ such that $\sigma(t^\star) = \sigma_{\text{ch}}$.

**Score Function and Reverse-Time Dynamics.** Let $\{\mathbf{x}_t\}_{t \in [0,T]}$ follow the forward SDE in (3), and let $p_t(\mathbf{x})$ be the resulting marginal density at time $t$. The *score* is the gradient of the log-density $\nabla_\mathbf{x} \log p_t(\mathbf{x})$. Score-based models learn a neural approximation $s_\theta(\mathbf{x}, t) \approx \nabla_\mathbf{x} \log p_t(\mathbf{x})$ from noisy samples at different noise levels. Given access to the score, one can construct a reverse-time stochastic process whose marginals evolve from a simple noise distribution at $t = T$ back to the data distribution at $t = 0$ (Song et al., 2020b):

$$d\mathbf{x}_t = \left[ f(\mathbf{x}_t, t) - g(t)^2 \nabla_\mathbf{x} \log p_t(\mathbf{x}_t) \right] dt + g(t)\,d\bar{\mathbf{w}}_t, \quad (5)$$

where $d\bar{\mathbf{w}}_t$ is a reverse-time Wiener increment. Instead of sampling the reverse SDE, one may integrate the probability flow ODE (PF-ODE), a deterministic dynamics that shares the same marginals $\{p_t\}$ as the SDE (Song et al., 2020b):

$$d\mathbf{x}_t = \left[ f(\mathbf{x}_t, t) - \tfrac{1}{2}g(t)^2 \nabla_\mathbf{x} \log p_t(\mathbf{x}_t) \right] dt. \qquad (6)$$

In practice, we replace the intractable score by a learned network $s_\theta(\mathbf{x}, t)$ and integrate the resulting reverse dynamics. In our setting we use the VE construction, where $f(\mathbf{x}, t) = 0$ and $g(t)^2 = \frac{d\sigma(t)^2}{dt}$, and we implement decoding by integrating the PF-ODE.

**Denoising Score Matching (DSM) Objective.** Score-based models are commonly trained via (denoising) score matching (Hyvärinen, 2005; Song et al., 2020b). Under the VE marginal (4) the conditional score admits a closed form (Song et al., 2020b):

$$\nabla_{\mathbf{x}_t} \log p(\mathbf{x}_t \mid \mathbf{x}_0) = -\frac{\mathbf{x}_t - \mathbf{x}_0}{\sigma(t)^2} = -\frac{\varepsilon}{\sigma(t)}. \qquad (7)$$

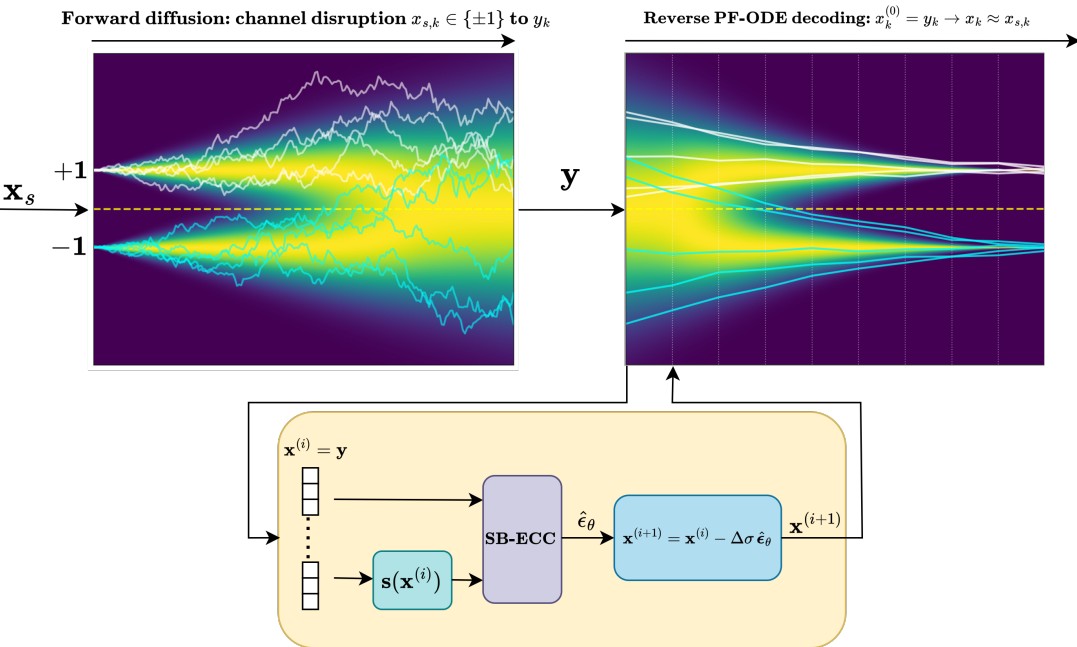

*Figure 1.* Per-bit view of decoding as deterministic denoising. **Left:** forward corruption (channel as a forward diffusion): each curve is one coordinate $k$, where a clean BPSK symbol $x_{s,k} \in \{\pm 1\}$ is perturbed into $y_k = x_{s,k} + z_k$ as the noise level increases. **Right:** reverse probability-flow ODE (PF-ODE) decoding: starting from $x_k^{(0)} = y_k$, the learned denoiser (with parity/ syndrome guidance) drives each coordinate toward a valid codeword, concentrating mass back near $\pm 1$. Background heatmaps depict empirical marginal density over coordinates, and the dashed line marks the hard-decision boundary at $0$.

A standard DSM objective is

$$\mathcal{L}_{\text{DSM}}(\theta) = \mathbb{E}_{t,\mathbf{x}_0,\boldsymbol{\varepsilon}}\Big[\lambda(t)\left\|s_\theta(\mathbf{x}_t,\, t) - \nabla_{\mathbf{x}_t}\log p(\mathbf{x}_t \mid \mathbf{x}_0)\right\|_2^2\Big],$$
$$(8)$$

which, using (7), is equivalent to

$$\mathcal{L}_{\text{DSM}}(\theta) = \mathbb{E}_{t,\mathbf{x}_0,\boldsymbol{\varepsilon}}\Big[\lambda(t)\left\|s_\theta(\mathbf{x}_t,\, t) + \frac{\boldsymbol{\varepsilon}}{\sigma(t)}\right\|_2^2\Big]. \quad (9)$$

In practice, many works (Ho et al., 2020; Dhariwal & Nichol, 2021; Karras et al., 2022) adopt equivalent parameterizations, e.g., predicting the noise $\boldsymbol{\varepsilon}$ (or $\mathbf{x}_0$) instead of the score, since under the VE marginal these are related by $s_\theta(\mathbf{x}_t, t) \approx -\hat{\boldsymbol{\varepsilon}}_\theta(\mathbf{x}_t, t)/\sigma(t)$ up to a time-dependent scaling (Song et al., 2020b).

## 4. Method

We propose a score-based decoding framework for binary linear block codes that operates directly on the full signed channel observation. Under BPSK over AWGN, the transmitted bit-codeword $\mathbf{x} \in \{0,1\}^n$ is mapped to $\mathbf{x}_s \in \{\pm 1\}^n$ (e.g., $\mathbf{x}_s = \mathbf{1} - 2\mathbf{x}$), and the receiver observes

$$\mathbf{y} = \mathbf{x}_s + \mathbf{z}, \qquad \mathbf{z} \sim \mathcal{N}(\mathbf{0}, \sigma_{\text{ch}}^2 I_n). \quad (10)$$

Many model-free decoders first preprocess $\mathbf{y}$ into a *reliability-only* representation using $|\mathbf{y}|$ together with parity

information derived from hard decisions (e.g., a syndrome) (Bennatan et al., 2018; Choukroun & Wolf, 2022b; Park et al., 2024). This pre-processing improves generalization in deep decoders by providing symmetry-consistent reliability features while injecting code constraints through hard-decision parity information (Bennatan et al., 2018). In a score-based formulation, however, the model is used as a *continuous* denoising/score field in $\mathbb{R}^n$ (to be integrated by an SDE/ODE solver), and the mapping $\mathbf{y} \mapsto |\mathbf{y}|$ is non-invertible: it collapses each equivalence class $\{\mathbf{s} \odot \mathbf{y} : \mathbf{s} \in \{\pm 1\}^n\}$ (size $2^n$) to the same input, i.e., $|\mathbf{s} \odot \mathbf{y}| = |\mathbf{y}|$ for all $\mathbf{s}$, discarding per-coordinate directional information. An overview of our formulation is shown in Figure 1, we interpret the channel observation as a forward VE perturbation (left) and decode by integrating the reverse PF-ODE in $\sigma$-space using the learned denoiser with parity guidance (right). In principle, parity cues such as a hard-decision syndrome can partially disambiguate these equivalence classes, and we do not claim that learning the score from $(|\mathbf{y}|, \mathbf{s}(\mathbf{y}))$ is impossible. Rather, we treat signed observations as a modeling choice that preserves geometry and quantify its impact empirically (Section 5.3).

**Transmission as forward diffusion.** Following the diffusion view of decoding (Choukroun & Wolf, 2022a), we align the AWGN observation with a VE perturbation pro-

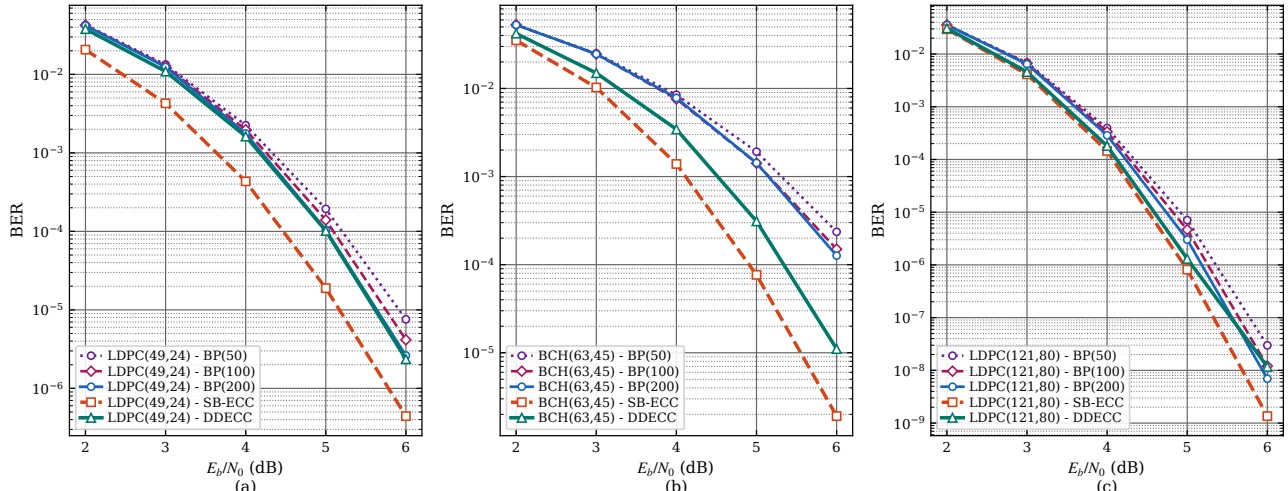

*Figure 2.* BER versus $E_b/N_0$ for representative code instances: (a) LDPC$(49, 24)$, (b) BCH$(63, 45)$, and (c) LDPC$(121, 80)$. The curves compare SB-ECC, DDECC, and BP decoding with 50, 100, and 200 iterations.

cess. Let $\mathbf{x}_0 \in \{\pm 1\}^n$ denote a clean BPSK codeword and define a noisy state $\mathbf{x}_t = \mathbf{x}_0 + \sigma(t)\boldsymbol{\epsilon}$, $\qquad \boldsymbol{\epsilon} \sim \mathcal{N}(\mathbf{0}, I_n)$, such that the channel observation corresponds to $\mathbf{y} \equiv \mathbf{x}_t$ at some (unknown) noise level $\sigma(t^\star) = \sigma_{\mathrm{ch}}$.

**Noise schedule.** We parameterize the VE noise level by $t \in [0, 1]$ using a simple linear schedule (Karras et al., 2022):

$$\sigma(t) = \sigma_{\min} + (\sigma_{\max} - \sigma_{\min})t. \tag{11}$$

Here, $\sigma_{\min}$ and $\sigma_{\max}$ are fixed hyperparameters that set the minimum and maximum noise levels used during training and sampling. This induces the Gaussian forward kernel $p(\mathbf{x}_t \mid \mathbf{x}_0) = \mathcal{N}(\mathbf{x}_t; \mathbf{x}_0, \sigma^2(t)I)$.

### 4.1. Model

We use the CrossMPT architecture (Park et al., 2024), which represents decoding as interactions between *variable-node (VN)* tokens and *check-node (CN)* tokens coupled through the Tanner graph. CrossMPT treats VN features and CN features (syndrome/parity information) as separate modalities and uses masked attention to restrict message exchange to edges indicated by the parity check matrix $H$. All CN features and Tanner-graph masking are identical to CrossMPT; only the VN observation channel is changed from magnitude-only to signed. Unlike reliability-based decoders that output bit logits or bit-flip probabilities, our network outputs a *continuous* denoising direction $\hat{\boldsymbol{\epsilon}}_\theta(\mathbf{y}, \mathbf{s}(\mathbf{y})) \in \mathbb{R}^n$

### 4.2. Training Objective

We train with the standard diffusion noise-parameterization (Ho et al., 2020). During training we sample a clean

BPSK codeword $\mathbf{x}_0 \in \{\pm 1\}^n$, draw $t \sim \mathcal{U}(0, 1)$ and $\boldsymbol{\epsilon} \sim \mathcal{N}(\mathbf{0}, I_n)$, and construct $\mathbf{y} = \mathbf{x}_0 + \sigma(t)\boldsymbol{\epsilon}$. We compute the corresponding syndrome $\mathbf{s}(\mathbf{y})$ and minimize the noise prediction loss

$$\mathcal{L}_\epsilon = \mathbb{E}\left[\|\hat{\boldsymbol{\epsilon}}_\theta(\mathbf{y}, \mathbf{s}(\mathbf{y})) - \boldsymbol{\epsilon}\|_2^2\right]. \tag{12}$$

This objective is equivalent (up to a known noise-level scaling) to denoising score matching for Gaussian perturbations (Song & Ermon, 2019; Song et al., 2020b), and yields a denoiser/score surrogate that can be used inside our iterative decoding dynamics (described next in Section 4.3).

### 4.3. Inference

Our inference process reverses the diffusion process to recover the clean BPSK codeword $\mathbf{x}_0 \in \{\pm 1\}^n$ from the received vector $\mathbf{y}$. We integrate the PF-ODE in *symbol space* (BPSK), and obtain decoded bits by $\hat{\mathbf{x}} = \mathrm{bin}(\mathrm{sign}(\mathbf{x}^{(N_{\mathrm{steps}})}))$ (or the early-stopped iterate). Following the VE formulation, this can be viewed as solving the probability flow ODE (PF-ODE) associated with the VE-SDE (Song et al., 2020b).

$$d\mathbf{x} = -\frac{1}{2}\frac{d[\sigma^2(t)]}{dt}\nabla_{\mathbf{x}}\log p_t(\mathbf{x})\,dt. \tag{13}$$

Using $d\sigma^2(t)/dt = 2\sigma(t)\dot{\sigma}(t)$ and the VE relation $\nabla_{\mathbf{x}}\log p_t(\mathbf{x}) \approx -\hat{\boldsymbol{\epsilon}}_\theta(\mathbf{x}, \mathbf{s}(\mathbf{x}))/\sigma(t)$ under Gaussian perturbations, we obtain an equivalent PF-ODE written in $\sigma$-space $d\mathbf{x} = \hat{\boldsymbol{\epsilon}}_\theta(\mathbf{x}, \mathbf{s}(\mathbf{x}))\,d\sigma$. We integrate from $\sigma_{\max}$ down to $\sigma_{\min}$ (hence $d\sigma < 0$), yielding a deterministic trajectory that progressively removes noise.

**Scheduling without time conditioning.** Standard diffusion samplers often discretize the *time* interval $t \in [0, 1]$

*Table 1.* Main decoding results across BCH, Polar, LDPC, MacKay, and CCSDS codes for BP, AR-BP (Nachmani & Wolf, 2021), CrossMPT (Park et al., 2024), DDECC (Choukroun & Wolf, 2022a), and SB-ECC. Results are reported as $-\ln(\text{BER})$ at $E_b/N_0 \in \{4, 5, 6\}$ dB, where higher is better. DDECC and SB-ECC use the same CrossMPT backbone for a controlled method-level comparison. BP and AR-BP results use 50 decoding iterations, except for gray AR-BP entries, which use 5 iterations. The best result is marked in **bold** and the second-best result is underlined.

| Architecture | | BP-based decoders | | | | | | Model-free decoders | | | | | | | | | |
| --- | --- | --- | --- | --- | --- | --- | --- | --- | --- | --- | --- | --- | --- | --- | --- | --- | --- |
| Code Type | Params | BP | | | AR BP | | | CrossMPT | | | DDECC | | | SB-ECC (Ours) | | |
| | | 4 | 5 | 6 | 4 | 5 | 6 | 4 | 5 | 6 | 4 | 5 | 6 | 4 | 5 | 6 |
| BCH | (63,36) | 4.03 | 5.42 | 7.26 | 4.57 | 6.39 | 8.92 | 5.03 | 6.91 | 9.37 | 5.30 | 7.32 | 10.25 | **5.74** | **8.12** | **11.20** |
| | (63,45) | 4.36 | 5.55 | 7.26 | 4.97 | 6.90 | 9.41 | 5.90 | 8.20 | 11.62 | 5.67 | 8.07 | 11.41 | **6.58** | **9.48** | **13.17** |
| | (63,51) | 4.50 | 5.82 | 7.42 | 5.17 | 7.16 | 9.53 | 5.78 | 8.08 | 11.41 | 5.37 | 7.48 | 10.51 | **6.19** | **8.82** | **12.45** |
| POLAR | (64,32) | 4.26 | 5.38 | 6.50 | 5.57 | 7.43 | 9.82 | 7.50 | 9.97 | 13.31 | 7.03 | 9.69 | 12.97 | **7.77** | **10.30** | **13.78** |
| | (64,48) | 4.74 | 5.94 | 7.42 | 5.41 | 7.19 | 9.30 | 6.51 | **8.70** | **11.31** | 6.07 | 8.40 | 10.90 | **6.63** | 8.64 | 11.27 |
| | (128,64) | 4.10 | 5.11 | 6.15 | 4.84 | 6.78 | 9.30 | 7.52 | 11.21 | 14.76 | 8.16 | 12.04 | 16.27 | **9.03** | **13.13** | **16.94** |
| | (128,86) | 4.49 | 5.65 | 6.97 | 5.39 | 7.37 | 10.13 | 7.51 | 10.83 | 15.24 | 7.92 | 11.34 | 15.61 | **8.03** | **11.48** | **16.10** |
| | (128,96) | 4.61 | 5.79 | 7.08 | 5.27 | 7.44 | 10.20 | 7.15 | 10.15 | 13.13 | 7.24 | **10.32** | 13.26 | **7.64** | 10.32 | **13.37** |
| LDPC | (49,24) | 6.23 | 8.19 | 11.72 | 6.58 | 9.39 | 12.39 | 6.68 | 9.52 | 13.19 | 6.26 | 9.07 | 12.71 | **7.74** | **10.88** | **14.63** |
| | (121,60) | 5.64 | 8.88 | 13.33 | 5.22 | 8.31 | 13.07 | 5.74 | 9.26 | 14.78 | 6.22 | 10.18 | 15.89 | **6.33** | **10.38** | **16.38** |
| | (121,70) | 6.91 | 10.66 | 15.62 | 6.45 | 10.01 | 14.77 | 7.06 | 11.39 | 17.52 | 7.64 | 12.30 | 17.98 | **7.75** | **12.69** | **19.24** |
| | (121,80) | 7.84 | 11.86 | 17.33 | 7.22 | 11.03 | 15.90 | 7.99 | 12.75 | 18.15 | 8.62 | 13.55 | 18.26 | **8.84** | **14.02** | **20.42** |
| MacKay | (96,48) | 6.84 | 9.40 | 12.57 | 7.43 | 10.65 | 14.65 | 7.97 | 11.77 | 15.52 | 8.58 | 12.48 | 16.04 | **8.97** | **12.82** | **16.25** |
| CCSDS | (128,64) | 8.00 | 12.11 | 17.17 | 7.25 | 10.99 | 16.36 | 7.68 | 11.88 | 17.50 | 8.40 | 13.18 | 17.47 | **8.60** | **13.43** | **19.20** |

using a fixed grid, which typically induces *non-uniform* increments in the corresponding noise level $\sigma(t)$. This is natural for time-conditional models that start from a known $t$ (usually $t = T = 1$). In channel decoding, however, the received vector $\mathbf{y} = \mathbf{x}_0 + \sigma^\star \mathbf{z}$ lies at an *unknown* intermediate noise level $\sigma^\star$, and our denoiser is not conditioned on $t$ (nor on $\sigma^\star$). To avoid locating $\mathbf{y}$ on a non-uniform $t$-grid, we discretize *directly* in $\sigma$.

**Uniform discretization in $\sigma$-space.** With the linear schedule in Eq. (11), a uniform grid $t_k = \frac{k}{N_{\text{steps}}}$ induces

$$\sigma_k = \sigma_{\min} + (\sigma_{\max} - \sigma_{\min})\frac{k}{N_{\text{steps}}},$$
$$\Delta\sigma = \frac{\sigma_{\max} - \sigma_{\min}}{N_{\text{steps}}}. \quad (14)$$

The hyperparameter $N_{\text{steps}}$ therefore controls the runtime–accuracy trade-off.

**Discrete solver and early stopping.** Starting from $\mathbf{x}^{(0)} = \mathbf{y}$, we apply Euler updates in $\sigma$-space:

$$\mathbf{x}^{(i+1)} = \mathbf{x}^{(i)} - \Delta\sigma\,\hat{\boldsymbol{\epsilon}}_\theta\left(\mathbf{x}^{(i)}, \mathbf{s}(\mathbf{x}^{(i)})\right) \quad (15)$$

We apply (15) $i = 0, \ldots, N_{\text{steps}} - 1$, after each step we form a hard decision and compute the syndrome; if $\mathbf{s}(\mathbf{x}^{(i)}) = \mathbf{0}$,

we terminate early and output the corresponding codeword. Otherwise, we continue up to the maximum budget $N_{\text{steps}}$. We also evaluate higher-order solvers, such as DPM-Solver (Lu et al., 2022), under the same $\Delta\sigma$ discretization to study performance–latency trade-offs in Section 5.2.

## 5. Experiments

We evaluate our score-based decoder on standard linear block code benchmarks and compare against strong neural and classical baselines. We also study inference-time trade-offs by varying the ODE solver and step budget.

### 5.1. Experimental Setup

**Codes and model configuration.** We follow common neural-decoding benchmarks and report results on Polar codes (Arikan, 2009), LDPC codes (Gallager, 1962), BCH codes (Bose & Ray-Chaudhuri, 1960), MacKay codes, and CCSDS codes. Unless stated otherwise, all model-free entries use the same backbone scale ($N = 6$ attention layers, $d = 128$ hidden dimension), matching the main CrossMPT configuration. For a fair method-level comparison with diffusion-based decoding, we retrain DDECC (Choukroun & Wolf, 2022a) using the same CrossMPT backbone as

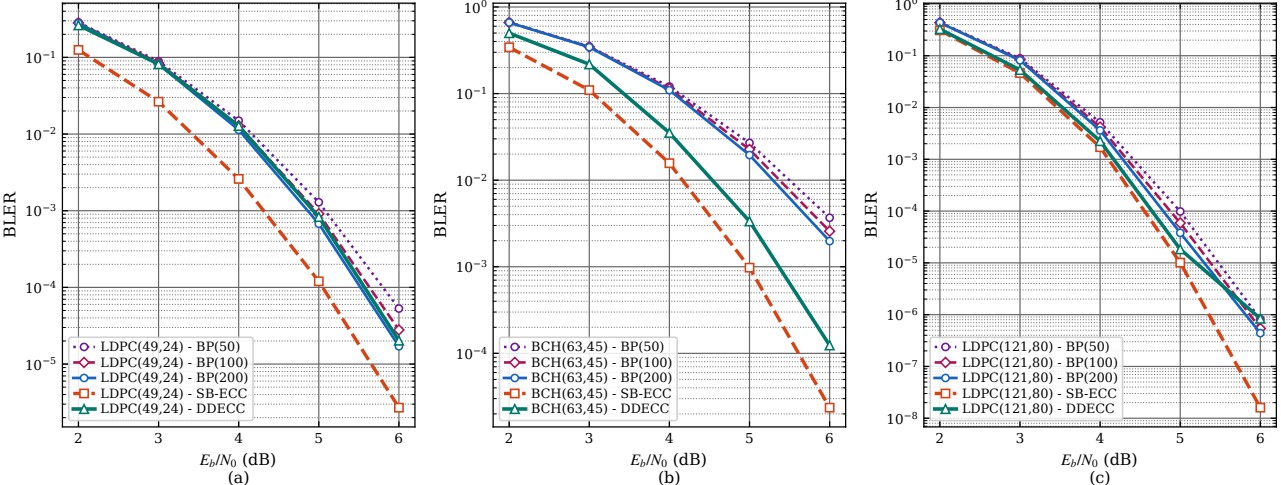

*Figure 3.* BLER versus $E_b/N_0$ for representative code instances: (a) LDPC$(49, 24)$, (b) BCH$(63, 45)$, and (c) LDPC$(121, 80)$. The curves compare SB-ECC, DDECC, and BP decoding with 50, 100, and 200 iterations.

SB-ECC.

**Training details.** All models are trained with Adam (Kingma & Ba, 2014) optimizer, learning rate $5 \times 10^{-4}$ and cosine annealing. We use batch size of 256 for codes with $n \leq 64$ and 128 for longer codes. We train for 1500 epochs with 1000 batches per epoch. Training samples are generated on-the-fly via BPSK over AWGN using a linear noise schedule $\sigma(t)$ with $\sigma_{\min} = 0.1$ and $\sigma_{\max} = 0.8$. Unless stated otherwise, inference uses Euler with $N_{\text{steps}} = 10$.

**Evaluation protocol and metric.** We report $-\ln(\text{BER})$ (higher is better) at $E_b/N_0 \in \{4, 5, 6\}$ dB. For each SNR, we decode until observing 500 frame errors or processing $10^8$ frames ($10^9$ for LDPC). When summarizing overall improvements, we also report SNR gain (dB), defined as the horizontal $E_b/N_0$ shift required for the strongest prior method to match our BER, computed by interpolation between the evaluated $E_b/N_0$ points.

**Baselines.** We compare against BP, AR-BP (Nachmani & Wolf, 2021), CrossMPT (Park et al., 2024), and DDECC (Choukroun & Wolf, 2022a). BP and AR-BP results are reported using 50 decoding iterations, except for the gray AR-BP entries, which correspond to the 5-iteration results reported by Nachmani & Wolf (2021). For a fair method-level comparison between diffusion-based decoders, DDECC is trained from scratch using the CrossMPT model as its backbone, instead of the original ECCT backbone.

**Main results.** Overall, treating decoding as score-based continuous denoising yields consistent improvements over prior methods across code families and rates. Table 1 summarizes the main results. SB-ECC achieves the best BER in

*Table 2.* Solver trade-offs on selected codes. Entries are $-\ln(\text{BER})$ at $E_b/N_0 \in \{4, 5, 6\}$ dB. $N_s$ is the maximum number of solver steps (see 4.3); inference uses early stopping.

| Code | Params | Euler | | | | DPM | | | | Speedup |
|------|--------|-------|---|---|---|-----|---|---|---|---------|
| | | $N_s$ | 4 | 5 | 6 | $N_s$ | 4 | 5 | 6 | % |
| BCH | (63,36) | 9 | 5.77 | 8.04 | 11.17 | 5 | 5.77 | 8.04 | 11.17 | 7.40 |
| | (64,32) | 7 | 7.76 | 10.39 | 13.72 | 5 | 7.76 | 10.39 | 13.72 | 9.95 |
| POLAR | (128,64) | 8 | 8.73 | 12.63 | 16.81 | 6 | 8.73 | 12.62 | 16.82 | 6.51 |
| | (128,86) | 7 | 7.83 | 11.05 | 14.84 | 5 | 7.83 | 11.04 | 14.85 | 7.49 |
| LDPC | (49,24) | 9 | 7.69 | 10.77 | 14.71 | 6 | 7.69 | 10.77 | 14.73 | 8.96 |
| | (121,60) | 9 | 6.33 | 10.40 | 16.01 | 6 | 6.32 | 10.38 | 16.02 | 12.82 |

39/42 entries and ties for the best result in one additional entry, with an average SNR gain of $\approx 0.17$ dB and a maximum gain of $0.46$ dB over the strongest competing baseline. The remaining entries are within a small margin of the best competing method. Figures 2 and 3 provide the corresponding BER and BLER curves for representative code instances. These plots complement the aggregate $-\ln(\text{BER})$ results in Table 1 and show the behavior across the evaluated $E_b/N_0$ range, including comparisons to stronger BP baselines with 100 and 200 iterations.

### 5.2. Inference Trade-offs: Solvers and Steps

Solver choice provides a direct latency–accuracy knob at test time, without retraining, we can reduce runtime while preserving error-rate performance. Table 2 compares Euler (Song et al., 2020b) and DPM (Lu et al., 2022) on representative codes, reporting $-\ln(\text{BER})$ at $E_b/N_0 \in \{4, 5, 6\}$ dB and end-to-end decoding time. We discretize uniformly in $\sigma$-space and denote by $N_s$ the *maximum* step budget (§4.3). With early stopping, trajectories often terminate before $N_s$, practical cost is governed by the realized number

*Table 3.* Ablation on noise-level conditioning. We report $-\ln(\text{BER})$ (higher is better) at $E_b/N_0 \in \{4,5,6\}$ dB for SB-ECC (ours), hard-syndrome conditioning, and predicted-time conditioning.

| Method | | SB-ECC (ours) | | | Hard-syndrome | | | Pred-time cond. | | |
|---|---|---|---|---|---|---|---|---|---|---|
| Code | Params | 4 | 5 | 6 | 4 | 5 | 6 | 4 | 5 | 6 |
| BCH | (63,36) | 5.74 | 8.12 | 11.20 | 5.71 | 8.04 | 11.00 | 5.68 | 7.90 | 11.07 |
| POLAR | (64,32) | 7.77 | 10.30 | 13.78 | 7.76 | 10.38 | 13.63 | 7.71 | 10.41 | 13.61 |
| | (128,64) | 9.03 | 13.13 | 16.81 | 8.64 | 12.47 | 16.27 | 8.40 | 12.26 | 16.63 |

*Table 4.* Signed vs. magnitude-only input.

| Code | Params | Signed $\mathbf{y}$ | | | Reliability $|\mathbf{y}|$ | | |
|---|---|---|---|---|---|---|---|
| | | 4 | 5 | 6 | 4 | 5 | 6 |
| BCH | (31,16) | 7.33 | 9.95 | 13.03 | 2.93 | 3.36 | 3.83 |
| Polar | (64,32) | 7.77 | 10.30 | 13.78 | 2.87 | 3.28 | 3.77 |
| LDPC | (49,24) | 7.74 | 10.88 | 14.63 | 2.84 | 3.24 | 3.72 |

of function evaluations. We measure runtime by decoding the same number of samples per SNR using the same checkpoint, batching, and GPU, and report time reduction as $(t_{\text{Euler}} - t_{\text{DPM}})/t_{\text{Euler}} \times 100\%$. Across all tested codes and SNRs, DPM attains essentially the same $-\ln(\text{BER})$ as Euler using a smaller step budget, translating into consistent runtime reductions. Together with early stopping, this yields a simple drop-in path to faster decoding without sacrificing accuracy.

### 5.3. Ablation Studies

**Time Conditioning.** We consider the practical setting where the decoder is given only the channel observation $\mathbf{y}$ and has *no access* to the channel SNR (or any other side information) at test time. We therefore study whether providing an auxiliary *noise-level proxy* inferred from $\mathbf{y}$ is beneficial by comparing three training variants of the same backbone.

**SB-ECC (ours)** is trained exactly as in (12) without providing any time/noise input. Hard-syndrome conditioning, inspired by DDECC, computes the hard-decision syndrome via $\mathbf{y}_b = \text{bin}(\text{sign}(\mathbf{y}))$ and $\mathbf{s}(\mathbf{y}) = H\mathbf{y}_b^\top$. We summarize it by the parity-error count $e(\mathbf{y}) \triangleq \sum_{i=1}^{n-k} s_i(\mathbf{y}) \in \{0, \ldots, n-k\}$, embed $e(\mathbf{y})$ with a learned embedding, and inject it by element-wise modulation of the initial token embeddings. Predicted-time conditioning adds a lightweight 2-layer MLP $\phi_\theta$ that predicts the normalized diffusion time $\hat{t} = \phi_\theta(\mathbf{y}, \mathbf{s}(\mathbf{y}))$, and conditions the decoder on $\hat{t}$. We train it jointly with $\mathcal{L}_{\text{joint}} = (1 - \gamma_{\text{time}})\mathcal{L}_\epsilon + \gamma_{\text{time}}\mathcal{L}_t$, where $\mathcal{L}_\epsilon$ is (12) and $\mathcal{L}_t = \|\hat{t} - t\|_2^2$, using $\gamma_{\text{time}} = 0.1$. Empirically, SB-ECC performs best (Table 3), suggesting that in our signed-input formulation the decoder can infer the effective noise regime from $\mathbf{y}$ directly. Conditioning on either a syndrome-based proxy or a learned time estimate does not improve performance. This observation is consistent with findings in denoising generative modeling that oracle noise conditioning is not always required and that additional proxy signals can sometimes hurt (Sun et al., 2025), and we find a similar effect in the error-correction setting.

**Signed vs. magnitude-only observations.** As shown in Table 4, to test whether discarding the sign harms learning the denoising vector field, we reran training with the *only* modification $\mathbf{y} \leftarrow |\mathbf{y}|$ at the network input, keeping the architecture, optimization, noise schedule, and data generation identical. Across $\text{BCH}(31, 16)$, $\text{Polar}(64, 32)$, and $\text{LDPC}(49, 24)$, the magnitude-only variant fails to learn: the training objective stagnates and decoding performance does not improve over the course of training. For all mentioned codes, the loss quickly plateaus at $\mathcal{L} \approx 5.0 \times 10^{-1}$, consistent with a near-trivial predictor.

Concretely, in the magnitude-only setting the network collapses to a near-zero noise estimator: $\hat{\epsilon}_\theta(\mathbf{y}, \mathbf{s}(\mathbf{y})) \approx \mathbf{0}$, with empirical statistics $\min \approx -0.1$, $\max \approx 0.1$, and mean $\approx 0$. As a result, the predicted clean estimate satisfies $\hat{\mathbf{x}}_0 \approx \mathbf{y}$, meaning that hard decisions are unchanged:

$$\text{sign}(\hat{\mathbf{x}}_0) = \text{sign}(\mathbf{x}_t) = \text{sign}(\mathbf{y}).$$

Therefore, the overall decoder reduces to uncoded hard-decision detection on the channel output, yielding essentially no coding gain.

## 6. Conclusion

We presented **SB-ECC**, a score-based decoder that casts soft decoding as continuous-time denoising via the PF-ODE, and learns a *continuous* additive-noise (score) field directly from the *raw signed* channel observation under parity constraints. Across 42 code/SNR settings, SB-ECC achieves the best BER in $39/42$ entries, with an average SNR gain of $\approx 0.17$ dB and a maximum gain of $0.46$ dB over the strongest competing baseline. The formulation also enables a practical latency–accuracy knob at inference: replacing Euler with a higher-order solver (DPM) preserves $-\ln(\text{BER})$ while reducing end-to-end decoding time by $8.86\%$ on average (up to $12.82\%$). Finally, our results indicate that learning decoding directly from raw observations can generalize to mid-length codes despite codeword-space sparsity, and that recent attention-based architectures make learning such continuous guidance fields feasible in practice.

## Impact Statement

This work aims to advance machine learning methods for soft decoding of error-correcting codes. Improved learned decoders may benefit communication and storage systems by increasing robustness and enabling flexible accuracy–latency trade-offs under fixed computational budgets. As with many general-purpose advances in communication technology, the same capabilities could be incorporated into both beneficial and potentially harmful applications depending on the deployment context. We encourage responsible use and consideration of relevant safeguards and regulations when integrating learned decoders into real-world systems.

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

*Table 5.* Iterations to reach zero syndrome (mean $\pm$ std) for Euler decoding with 10 vs. 5 steps, and the corresponding deltas. $\Delta(-\ln\text{BER})$ is computed as (5 steps) $-$ (10 steps), so positive values indicate improved BER.

| Code | $E_b/N_0$ | #It (10 steps) | #It (5 steps) | $\Delta$#It | $\Delta(-\ln\text{BER})$ |
|---|---|---|---|---|---|
| BCH (63,36) | 4 | 2.92$\pm$1.82 | 1.68$\pm$0.87 | -1.24 | -0.09 |
| | 5 | 1.85$\pm$1.31 | 1.16$\pm$0.71 | -0.70 | -0.10 |
| | 6 | 1.09$\pm$1.02 | 0.75$\pm$0.62 | -0.34 | -0.15 |
| BCH (63,45) | 4 | 1.95$\pm$1.53 | 1.20$\pm$0.80 | -0.75 | -0.05 |
| | 5 | 1.12$\pm$1.06 | 0.76$\pm$0.64 | -0.36 | -0.10 |
| | 6 | 0.58$\pm$0.78 | 0.44$\pm$0.54 | -0.14 | -0.12 |
| BCH (63,51) | 4 | 1.55$\pm$1.51 | 1.00$\pm$0.81 | -0.56 | -0.02 |
| | 5 | 0.80$\pm$0.94 | 0.58$\pm$0.60 | -0.22 | -0.05 |
| | 6 | 0.38$\pm$0.64 | 0.30$\pm$0.47 | -0.07 | -0.08 |
| LDPC (49,24) | 4 | 3.11$\pm$1.53 | 1.79$\pm$0.81 | -1.31 | -0.07 |
| | 5 | 2.14$\pm$1.37 | 1.30$\pm$0.76 | -0.84 | -0.11 |
| | 6 | 1.34$\pm$1.17 | 0.88$\pm$0.69 | -0.47 | -0.05 |
| LDPC (121,60) | 4 | 3.77$\pm$1.23 | 2.12$\pm$0.62 | -1.65 | -0.03 |
| | 5 | 2.76$\pm$0.97 | 1.64$\pm$0.55 | -1.13 | -0.05 |
| | 6 | 1.94$\pm$0.92 | 1.21$\pm$0.53 | -0.73 | -0.05 |
| POLAR (64,48) | 4 | 1.80$\pm$1.32 | 1.15$\pm$0.77 | -0.65 | 0.01 |
| | 5 | 1.03$\pm$1.04 | 0.71$\pm$0.64 | -0.32 | -0.00 |
| | 6 | 0.51$\pm$0.76 | 0.39$\pm$0.53 | -0.12 | -0.01 |
| POLAR (128,64) | 4 | 4.20$\pm$1.14 | 2.35$\pm$0.62 | -1.85 | -0.10 |
| | 5 | 3.12$\pm$1.07 | 1.81$\pm$0.59 | -1.31 | -0.07 |
| | 6 | 2.18$\pm$1.03 | 1.34$\pm$0.58 | -0.84 | -0.02 |
| POLAR (128,86) | 4 | 2.70$\pm$1.07 | 1.61$\pm$0.61 | -1.09 | -0.06 |
| | 5 | 1.83$\pm$0.95 | 1.16$\pm$0.54 | -0.67 | -0.11 |
| | 6 | 1.11$\pm$0.86 | 0.79$\pm$0.52 | -0.32 | -0.08 |
| POLAR (128,96) | 4 | 2.32$\pm$1.13 | 1.42$\pm$0.65 | -0.90 | -0.03 |
| | 5 | 1.47$\pm$0.93 | 0.98$\pm$0.53 | -0.50 | -0.05 |
| | 6 | 0.82$\pm$0.79 | 0.63$\pm$0.53 | -0.20 | -0.01 |

# 7. Appendix

## 7.1. Improvement Measured in dB

Figure 4 presents BER curves for representative code instances with large SNR gains of SB-ECC relative to the strongest competing baseline at matched BER. We quantify improvement using *SNR gain*, defined as the reduction in $E_b/N_0$ required by SB-ECC to achieve the same BER as the best competing baseline. Across the three cases shown in Figure 4, the gains at $E_b/N_0 = 5$ dB are $0.37$ dB for BCH$(63, 45)$, $0.37$ dB for LDPC$(49, 24)$, and $0.27$ dB for BCH$(63, 36)$.

## 7.2. Early Stopping Behavior and Effective Compute

Table 5 measures how quickly Euler decoding reaches a valid codeword under the parity constraints. At each solver iteration $i$, we convert the current continuous estimate into bits, $\hat{\mathbf{x}}^{(i)} = \text{bin}(\text{sign}(\mathbf{x}^{(i)}))$, and compute its syndrome. The reported #It is the first iteration index at which the syndrome becomes zero, and the table summarizes this stopping iteration by its mean and standard deviation. This metric is reported alongside BER because the two capture different aspects of decoding. The stopping iteration reflects how fast the solver finds a parity-consistent candidate, while BER

evaluates whether that candidate matches the transmitted codeword. Together, they quantify the effectiveness of early stopping and its impact on accuracy.

Across all code families and SNRs, parity satisfaction occurs in only a few iterations on average. Even when allowing a maximum of 10 solver steps, the mean stopping iteration is typically in the 1-4 range (with decreasing iterations as $E_b/N_0$ increases), indicating that the PF-ODE trajectory quickly moves into the code manifold. Reducing the solver budget to 5 steps further decreases the stopping iteration (negative $\Delta$#It), as expected given the smaller iteration horizon.

Importantly, this analysis also clarifies the *latency gap* to one-step decoders. A one-step model corresponds to a single neural evaluation, whereas our method performs multiple evaluations. However, the measured mean iteration counts show that we are not an order of magnitude slower in typical operating regimes: in the mean case we require only a small constant number of updates to reach a valid codeword. This modest increase in compute yields substantial BER improvements over one-step baselines (see main results), highlighting a favorable latency-accuracy trade-off. Several additional denoiser evaluations are sufficient to capture most of the performance gains, while still keeping inference close to one-step latency in practice.

## 7.3. Scalability to Longer Codes

To further evaluate scalability beyond the block lengths used in the main benchmark, we include additional experiments on longer codes.

Table 6 reports $-\ln(\text{BER})$ at $E_b/N_0 \in \{4, 5, 6\}$ dB for LDPC $(204, 102)$ and LDPC $(529, 440)$. BP is evaluated with 50 decoding iterations. The results suggest that SB-ECC remains competitive on longer codes.

*Table 6.* Longer LDPC-code results. Values are reported as $-\ln(\text{BER})$ at $E_b/N_0 \in \{4, 5, 6\}$ dB, where higher is better. BP uses 50 decoding iterations.

| Code | BP-50 | | | SB-ECC | | |
|---|---|---|---|---|---|---|
| | 4 | 5 | 6 | 4 | 5 | 6 |
| LDPC $(204, 102)$ | 10.27 | 13.79 | 16.29 | **11.20** | **15.78** | **18.77** |
| LDPC $(529, 440)$ | **8.18** | 14.88 | 19.74 | 7.93 | **15.23** | **21.70** |

## 7.4. Robustness under Rayleigh Fading Channel

The main experiments focus on the standard AWGN setting. Following the Rayleigh-channel evaluation protocol used in ECCT(Choukroun & Wolf, 2022b) and CrossMPT(Park et al., 2024), we additionally evaluate SB-ECC under a Rayleigh fading channel. In this setting, the received signal is

$$\mathbf{y} = \mathbf{h} \odot \mathbf{x}_s + \mathbf{z}, \tag{16}$$

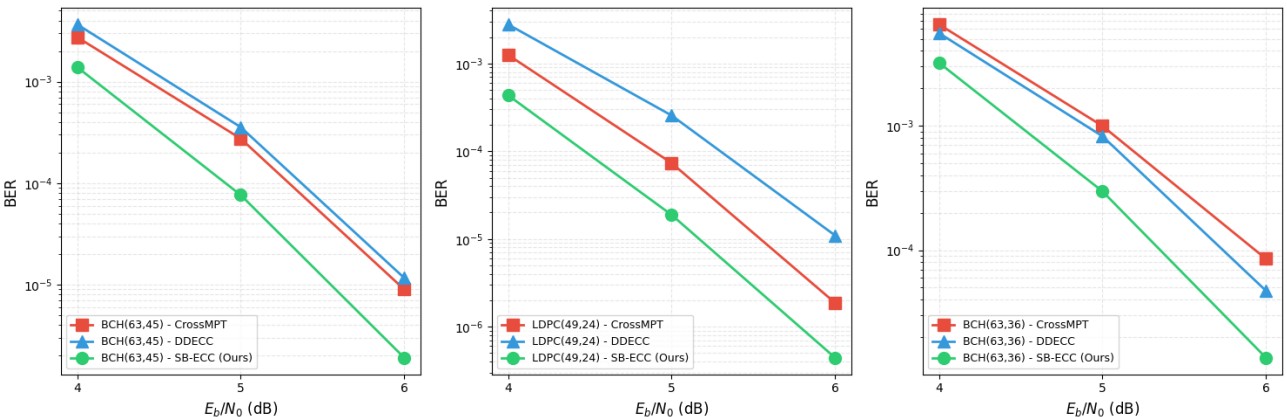

*Figure 4.* BER versus $E_b/N_0$ for three representative code instances with large SNR gains of SB-ECC (green). Left to right: BCH$(63, 45)$, LDPC$(49, 24)$, and BCH$(63, 36)$. Markers show simulated operating points at $E_b/N_0 \in \{4, 5, 6\}$ dB (lines connect points). SNR gain is defined as the reduction in $E_b/N_0$ required by SB-ECC to match the BER of the strongest competing baseline, computed via linear interpolation in the $-\ln(\text{BER})$ domain.

*Table 7.* Rayleigh-channel inference results using models trained under AWGN. Values are reported as $-\ln(\text{BER})$; higher is better. Best results are marked in bold.

| Code | Params | $E_b/N_0$ | BP-50 | BP-200 | CrossMPT | SB-ECC |
|---|---|---|---|---|---|---|
| LDPC | $(121, 70)$ | 4 | 4.06 | 4.26 | 4.25 | **4.62** |
| | | 5 | 5.10 | 5.48 | 5.53 | **6.02** |
| | | 6 | 6.34 | 6.89 | 7.11 | **7.92** |
| BCH | $(31, 16)$ | 4 | 3.79 | 3.99 | 5.53 | **5.74** |
| | | 5 | 4.36 | 4.67 | 6.55 | **6.86** |
| | | 6 | 4.93 | 5.37 | 7.61 | **8.07** |
| CCSDS | $(128, 64)$ | 4 | 5.25 | 5.59 | 5.25 | **5.77** |
| | | 5 | 6.78 | 7.37 | 6.94 | **7.70** |
| | | 6 | 8.72 | 9.64 | 8.92 | **10.08** |

where **h** is an i.i.d. Rayleigh-distributed fading vector with scale parameter $\alpha = 1$, $\odot$ denotes element-wise multiplication, and $\mathbf{z} \sim \mathcal{N}(\mathbf{0}, \sigma^2 I)$. Importantly, we use the same models trained under AWGN and evaluate them on Rayleigh samples without retraining or fine-tuning.

Table 7 reports $-\ln(\text{BER})$ at $E_b/N_0 \in \{4, 5, 6\}$ dB for BP with 50 and 200 iterations, CrossMPT, and SB-ECC. Across the evaluated code instances, SB-ECC achieves the best result among the available baselines, providing initial evidence that the learned score-based decoder retains useful robustness under a channel shift from AWGN training to Rayleigh inference.

### 7.5. Comparison with SCL Decoding

We compare SB-ECC with successive cancellation list (SCL) decoding on Polar codes. SCL is a classical Polar-specific decoder and therefore provides a strong specialized reference point for this code family. Table 8 reports $-\ln(\text{BER})$ at $E_b/N_0 \in \{4, 5, 6\}$ dB. The SCL and CrossMPT values are taken from Park et al. (2024), and we add the corresponding SB-ECC results. As expected, the

Polar-specific SCL decoder, especially with list size $L = 4$, remains a very strong baseline and achieves the best result in many settings.

*Table 8.* Comparison with SCL decoding on Polar codes. Results are reported as $-\ln(\text{BER})$. Higher is better; best results are marked in bold.

| Code | $E_b/N_0$ | SCL-1 | SCL-4 | Cross-MPT | SB-ECC |
|---|---|---|---|---|---|
| Polar $(64, 32)$ | 4 | 7.30 | **8.11** | 7.50 | 7.77 |
| | 5 | 9.67 | **10.70** | 9.97 | 10.30 |
| | 6 | 13.18 | **14.04** | 13.31 | 13.78 |
| Polar $(64, 48)$ | 4 | 6.19 | **6.69** | 6.51 | 6.63 |
| | 5 | 8.41 | 8.63 | **8.70** | 8.64 |
| | 6 | 10.97 | 11.24 | **11.31** | 11.27 |
| Polar $(128, 64)$ | 4 | 8.37 | **9.60** | 7.52 | 9.03 |
| | 5 | 11.69 | **13.16** | 11.21 | 13.13 |
| | 6 | 13.70 | **17.42** | 14.76 | 16.94 |
| Polar $(128, 86)$ | 4 | 7.54 | **9.26** | 7.86 | 8.03 |
| | 5 | 10.74 | **13.04** | 11.45 | 11.48 |
| | 6 | 15.14 | **17.13** | 15.47 | 16.10 |
| Polar $(128, 96)$ | 4 | 6.74 | **8.02** | 7.15 | 7.64 |
| | 5 | 9.53 | **11.60** | 10.15 | 10.32 |
| | 6 | 13.53 | **18.16** | 13.13 | 13.37 |

### 7.6. Latency and Throughput Compared to External Baselines

We complement the internal solver comparison in Section 5.2 with a direct runtime comparison against external baselines. Since CrossMPT is a one-step decoder, while SB-ECC and DDECC are iterative decoding methods, this comparison should be interpreted as a practical latency–accuracy trade-off rather than as a pure architecture comparison. All measurements are performed under the same hardware setting. For DDECC, we use the public repository implementation without adding an early-stopping rule.

Table 9 reports latency and throughput on BCH $(63, 45)$

at $E_b/N_0 \in \{3, 4, 5\}$ dB. As expected, the one-step CrossMPT decoder has the lowest latency. SB-ECC requires multiple denoising evaluations, but is substantially faster than the evaluated DDECC implementation across all tested SNRs. We note, however, that this runtime gap is partly implementation-dependent; adding a parity-check-based early-stopping criterion to DDECC would likely reduce its latency and increase its throughput, and may therefore move its practical runtime closer to SB-ECC.

*Table 9.* Latency and throughput comparison on BCH $(63, 45)$. Latency is reported in milliseconds and throughput in samples/sec.

| Method | Latency (ms) | | | Throughput | | |
|---|---|---|---|---|---|---|
| | 3 dB | 4 dB | 5 dB | 3 dB | 4 dB | 5 dB |
| SB-ECC | 49.96 | 30.76 | 18.34 | 20.02 | 32.51 | 54.53 |
| CrossMPT | **14.69** | **14.55** | **14.48** | **68.08** | **68.71** | **69.04** |
| DDECC | 266.06 | 266.85 | 265.10 | 3.76 | 3.75 | 3.77 |

