# OpenReview forum: "Score Based Error Correcting Code Decoder"
_ICML.cc/2026/Conference — ICML 2026 regular_

### Official Review · Reviewer_ZxXf · 2026-03-05

**Soundness:** 3
**Presentation:** 4
**Significance:** 3
**Originality:** 3
**Overall Recommendation:** 4
**Confidence:** 4

**Summary:**

This paper reinterprets error-correcting code decoding as score-based continuous denoising and Probability Flow ODE (PF-ODE) integration. Instead of the magnitude-only input $|y|$ typically used by existing model-free decoders, it directly utilizes the signed observation $y$. Furthermore, the model is trained across multiple noise levels without time/SNR conditioning, and it provides an inference budget-accuracy trade-off by employing Euler or DPM solvers. Consequently, the authors claim to achieve the best Bit Error Rate (BER) in 39 out of 42 code/SNR settings, with an average improvement of 0.21 dB and a maximum improvement of 0.44 dB.

**Compliance With Llm Reviewing Policy:**

Affirmed.

**Key Questions For Authors:**

1. On variable block lengths generalization and recent baselines: While the paper implies an ability to generalize to variable block lengths, the empirical evidence supporting this claim seems somewhat limited. Could the authors provide additional experimental results or clarify how well the model maintains its performance on unseen block lengths? Furthermore, how does the proposed method compare against the more recent foundation-model or hybrid variants mentioned in the related work, particularly in these generalization scenarios?

2. The current wall-clock runtime analysis primarily focuses on internal trade-offs (e.g., Euler vs. DPM solvers). Given that the continuous denoising approach requires iterative steps—even with early stopping—could the authors provide a direct latency and throughput comparison against state-of-the-art external baselines (including strong one-step decoders) under identical hardware settings? Providing a latency-accuracy Pareto front would greatly help clarify the practical viability of this approach.

**Limitations:**

1. While SB-ECC reports strong BER gains across many code/SNR settings, it is still unclear how well a single trained model transfers to substantially different regimes (e.g., new block lengths, code constructions outside the training set, or mismatched channel models), and how robust it is under real-world non-idealities (imperfect CSI, quantization, synchronization errors).

2. he approach relies on iterative score-based updates (Euler/DPM solvers) with early stopping, but the paper provides limited end-to-end comparisons against strong modern decoders in terms of latency/throughput/energy for a target BER. This leaves open whether the BER improvements translate into a better operating point under practical compute budgets.

**Strengths And Weaknesses:**

Strengths
•	Elegant Problem Reformulation: The motivation is highly convincing—specifically, the argument that folding the observation space into $|y|$ discards critical directional information necessary for learning a continuous guidance field.
•	Solid Empirical Evaluation: The experiments are robust. The authors provide comprehensive comparisons across multiple code families (e.g., BCH, Polar, LDPC) and effectively demonstrate solver trade-offs.
•	Strong Ablation Studies: The ablations clearly show that adding noise/time conditioning yields negligible benefits, and that reverting from signed inputs to magnitude-only inputs essentially leads to training collapse.
•	Practical Computational Cost: The argument that the computational overhead is not excessively high compared to one-step decoders, thanks to early stopping, is reasonably persuasive.

Weaknesses
•	Incremental Methodological Novelty: The core novelty lies in the input/decoding formulation rather than a fundamentally new architecture. Because it utilizes the existing CrossMPT backbone and simply applies signed-input and score-based training on top of it, the methodological leap is arguably somewhat limited.
•	Lack of Recent Baselines: While the related work section broadly discusses newer variants (e.g., foundation models, hybrid variants), the main empirical comparisons heavily focus on older or standard baselines like BP, AR-BP, CrossMPT, and DDECC. For an ICML-tier submission, more extensive comparisons with recent, state-of-the-art baselines are expected.
•	Insufficient Proof of Generalization: The paper implies "variable block lengths generalization" capabilities, but the experiments supporting this claim are not strong enough to fully validate it.
•	Limited External Runtime Trade-off Analysis: The wall-clock runtime comparisons are mostly internal (e.g., Euler vs. DPM), making the latency/fidelity trade-off analysis relatively weak when compared against external baselines.

---

> ### Author Rebuttal · Authors · 2026-03-31
>
> We thank the reviewer for the careful reading and constructive feedback.
>
> **Question 1 - Generalization across block lengths.**
>
> The wording around ``variable block lengths generalization'' should be made more precise. The paper does not consider the foundation-model setting in which a
> single trained model generalizes across unseen codes or unseen block lengths. Rather, the intended claim is that the proposed score-based decoding formulation performs well across multiple code families and block lengths when trained and evaluated on each setting, which is also the standard evaluation setting used by prior model-free neural decoders. To strengthen the empirical evidence within that scope, we are adding longer-code experiments, including LDPC$(204,102)$, LDPC$(529,440)$, and POLAR$(512,384)$, and will make this distinction explicit in the revision.
>
> **Question 1 - Generalization across block lengths.**
>
> The current runtime discussion emphasizes internal solver trade-offs, but the more informative practical comparison is against external baselines under matched hardware conditions. In response, we are adding direct latency and throughput comparisons against CrossMPT and DDECC under identical hardware settings. Since CrossMPT is a one-step decoder and DDECC is an iterative diffusion-based decoder on the same backbone, this controlled same-backbone comparison isolates the effect of the proposed score-based decoding formulation without conflating it with architectural differences.
>
> For BCH$(63,45)$, the measured latency and throughput are:
>
> | Method | Latency 3 dB (ms) | Latency 4 dB (ms) | Latency 5 dB (ms) | Throughput 3 dB (samples/s) | Throughput 4 dB (samples/s) | Throughput 5 dB (samples/s) |
> |---|---:|---:|---:|---:|---:|---:|
> | SB-ECC | 49.96 | 30.76 | 18.34 | 20.02 | 32.51 | 54.53 |
> | CrossMPT | 14.69 | 14.55 | 14.48 | 68.08 | 68.71 | 69.04 |
> | DDECC | 266.06 | 266.85 | 265.10 | 3.76 | 3.75 | 3.77 |
>
> As expected, the one-step CrossMPT baseline has the lowest latency, while SB-ECC is substantially faster than DDECC in this controlled comparison while delivering the accuracy gains discussed in the paper. For clarity, the DDECC numbers are measured using the public repository implementation without early stopping.
>
> We also agree that presenting the results through a latency-accuracy Pareto view would further clarify the practical operating trade-off. In the revised manuscript, we will make this trade-off more explicit by combining the added latency/throughput measurements with decoding performance across multiple codes and operating points. Within the scope of this paper, however, we believe that the controlled comparison to one-step CrossMPT and same-backbone DDECC is the most direct and fair way to evaluate the practical cost of the proposed score-based formulation itself.
>
> **Limitation 1**
>
> We thank the reviewer for raising this point. As noted above, transfer of a single trained model across unseen codes or block lengths is not the setting studied in this paper. On the channel side, however, we can broaden the evidence by adding results under a Rayleigh channel, which show that the proposed framework is not specific to AWGN.
> **Performance under Rayleigh channel.** We report $-\ln(\mathrm{BER})$ at $E_b/N_0 \in \{4,5,6\}$\,dB for BP-SPA, CrossMPT, and SB-ECC.
>
> | Code Type | Code Params | BP (50) 4 dB | BP (50) 5 dB | BP (50) 6 dB | BP (200) 4 dB | BP (200) 5 dB | BP (200) 6 dB | CrossMPT 4 dB | CrossMPT 5 dB | CrossMPT 6 dB | SB-ECC 4 dB | SB-ECC 5 dB | SB-ECC 6 dB |
> |---|---:|---:|---:|---:|---:|---:|---:|---:|---:|---:|---:|---:|---:|
> | LDPC | (121,70) | 4.06 | 5.10 | 6.34 | 4.259 | 5.48 | 6.89 | 4.25 | 5.53 | 7.11 | 4.62 | 6.02 | 7.92 |
> | BCH | (31,16) | 3.79 | 4.36 | 4.93 | 3.99 | 4.67 | 5.37 | 5.53 | 6.55 | 7.61 | 5.74 | 6.86 | 8.07 |
> | CCSDS | (128,64) | 5.25 | 6.78 | 8.72 | 5.59| 7.37| -- | 5.25 | 6.94 | 8.92 | 5.77 | 7.70 | 10.08|
>
> **Limitation 2 - Performance for target BER**
>
> We agree that the most informative practical comparison is not BER or runtime in isolation, but the operating point needed to reach a target BER/BLER. Measuring latency and throughput at fixed BER/BLER targets, together with a latency--accuracy Pareto view, would give a clearer end-to-end picture under realistic compute budgets.
>
> We were not able to complete this analysis within the rebuttal window, as it requires a substantially broader set of experiments across methods, SNRs, and stopping configurations. We do, however, view this as a high-priority addition. We will add it to the rebuttal if the runs are completed in time, and otherwise include it in the revised manuscript. This will make the practical trade-off of SB-ECC much clearer relative to one-step decoders and other strong baselines.

---

> > ### Author Rebuttal · Reviewer_ZxXf · 2026-04-04
> >
> > I think that the rebuttal reflects my concerns well.

---

> > > ### Author Response · Authors · 2026-04-06
> > >
> > > Thank you for the acknowledgement. We appreciate your positive assessment.

---

### Official Review · Reviewer_LjXg · 2026-03-05

**Soundness:** 3
**Presentation:** 3
**Significance:** 2
**Originality:** 2
**Overall Recommendation:** 4
**Confidence:** 4

**Summary:**

The authors propose SB-ECC, a score-based decoder for linear block codes that formulates soft decoding as continuous-time denoising process. The method uses a message-passing transformer backbone (CrossMPT) operating on signed channel observations. The authors first provide a thorough literature review of ML techniques applied to the decoding of error correcting codes. Then explanation on general coding theory, and diffusion process. The authors compare their proposed model with conventional literature benchmark: standard Belief Propagation (BP) (non ML benchmark), Neural Belief Propagation (NBP) (model based ML decoder), and CrossMPT or DDECC (model free ML decoders).

**Compliance With Llm Reviewing Policy:**

Affirmed.

**Final Justification:**

This paper proposes a score-based decoder for error correcting codes that formulates decoding as a denoising process using a message-passing transformer backbone (CrossMPT). The theoretical framework is sound, the writing is clear, and the experimental evaluation covers multiple code families.

My initial concerns centered on:
- The fairness of comparisons with standard BP decoders, particularly the use of an insufficient number of decoding iterations for LDPC codes and/or the absence of a computational complexity analysis.
- The use of unconventional performance metric and the absence of BLER results.
- Insufficient ablation studies, architectural details, and distinction between SB-ECC and DDECC.

The authors' rebuttal satisfactorily addressed these concerns. The addition of stronger BP baselines with 50, 100, and 200 iterations, provided in a timely manner during the rebuttal period, effectively addresses the concern about potential baseline underestimation for LDPC codes. The commitment to providing BER and BLER curves in the revised manuscript will facilitate comparison with published baselines. The commitment to provide controlled ablations and a clearer presentation of the CrossMPT backbone, as well as a better distinction between SB-ECC and DDECC will improve reproducibility. The latency and throughput comparison partially addresses the complexity concern; while a direct comparison with BP complexity would have been more informative, this is a reasonable scope limitation for a work focused on model-free approaches.

Conditional on the discussed manuscript updates, the rebuttal has addressed my main concerns and reinforced my positive assessment. I update my soundness score to "good" and maintain my recommendation of **"Weak Accept"**, acknowledging the incremental nature of the contribution and remaining open questions on practical complexity leave some room for further strengthening.

I thank the authors for their thorough and constructive responses.

**Key Questions For Authors:**

My main concern lies with the comparison with standard BP algorithms. I acknowledge that the authors use the same baseline from other papers, but I am not entirely convinced these are properly evaluated baselines. While an improvement is to be expected for Polar and BCH codes (since BP is not well suited for decoding such codes), the proof supporting the improvement provided by the proposed method in the case of LDPC codes should be more solid, since BP decoders when properly configured are well suited for LDPC decoding. In the present case, I understand that except for the LDPC (49,24) code, only 5 decoding iterations are considered. This is very small for LDPC codes and might very well hinder the decoding performance. Evidence from published literature shows that for CCSDS (128,64) with 15 iterations, -ln(BER @5dB)=10.83 compared to the -ln(BER @5dB)=9.65 listed in the paper under 5 iterations. From other works, unfortunately expressed in BLER and thus not directly comparable, I posit that the decoding of CCSDS under plain BP using 200 iterations would surpass the listed SB-ECC performance of 13.43 @5dB (Liva et al. https://arxiv.org/pdf/1610.00873). One could argue that 200 iterations of BP is more complex, which is true, but then for the comparison to be fair we should compare this complexity to that of SB-ECC. These elements do not change the conclusion regarding the performance comparison between SB-ECC and DDECC, but raise significant concerns as to the relevance of the approach with respect to standard methods.

- 1.	Can you provide BLER performance for SB-ECC on all evaluated codes, particularly LDPC codes? This would enable direct comparison with published BP baselines, including CCSDS (128,64) BP with 200 iterations from Liva et al. (https://arxiv.org/pdf/1610.00873). Additionally, can you explicitly specify the number of BP iterations used for all baselines in your experiments? If indeed 5 iterations were used for most LDPC codes, can you provide BP results with 50, 100, and 200 iterations to establish a fair comparison? Alternatively, if the claim is that SB-ECC achieves better performance at comparable computational cost, can you provide a detailed complexity analysis comparing the effective number of operations between SB-ECC and BP with different iteration counts? This information would significantly impact my evaluation, as it would clarify whether the observed improvements represent genuine algorithmic advantages or artifacts of under-configured baselines.

- 2.	Can you provide comprehensive ablation studies to isolate the sources of performance gains over DDECC? Specifically, it remains unclear whether improvements stem from the continuous-time diffusion formulation, the signed input representation, the CrossMPT backbone architecture, or a combination of these factors. Ablations comparing DDECC with signed input, DDECC with CrossMPT backbone, and SB-ECC with magnitude input (properly trained) would help decompose the contribution and clarify which design choices are most impactful. Understanding the individual contributions of these components would strengthen the paper's claims.

- 3.	How does SB-ECC compare with joint encoder-decoder learning techniques for error correction? The current work focuses on decoder learning with fixed codes, but learning both encoder and decoder jointly represents another promising venue for discovering high-performing ECC approaches, as demonstrated in recent works. Could you provide context on how your approach relates to or compares with these code learning techniques? While not critical to the current contribution, this discussion would help position your work within the broader landscape of learning-based error correction methods.

**Limitations:**

Yes.

**Strengths And Weaknesses:**

I start by providing a general appreciation of the paper's **strengths**. The paper presents a well-motivated approach to error correction decoding by adapting score-based diffusion models to this domain. The theoretical framework is sound, with the score-based formulation via probability-flow ODE properly derived and clearly presented. The experimental evaluation is comprehensive, covering multiple code families and demonstrating competitive performance compared to recent model-free approaches under similar evaluation conditions. The authors demonstrate good scientific practice by providing code in the supplementary material. The writing is clear with good structure and comprehensive literature review, making the main ideas effectively communicated. The problem addressed is important with broad practical relevance to communication systems.

I now provide an appreciation of the paper's **weaknesses**. The most critical issue concerns the fairness of comparisons with conventional BP decoders, where the experimental setup appears to systematically disadvantage the baseline by using insufficient iterations (5 iterations for most LDPC codes). This raises serious concerns about whether reported gains reflect algorithmic improvements or merely suboptimal baseline configurations. The absence of computational complexity analysis is problematic, as the transformer-based architecture likely requires more operations than BP. The backbone architecture is under-specified, relying heavily on external references. The paper lacks comprehensive ablation studies to isolate sources of performance gains over DDECC. Additionally, only BER is reported while BLER would enable comparison with published baselines. The contribution is incremental, building on existing work, and the practical advantage remains unclear without proper baseline evaluation and complexity analysis.

Now regarding the four review items:

**Soundness:** The theoretical formulation is sound and well-presented. The score-based framework via probability-flow ODE is properly derived, and the mathematical exposition is clear. However, the experimental evaluation appears to have important weaknesses:

- The decoding conditions, particularly the number of BP iterations, are not clearly stated in the paper and must be inferred from cited references. This is a critical parameter that should be explicitly specified. Based on Nachmani & Wolf 2021, the BP baseline appears to use only 5 iterations for all codes except the LDPC (49,24) code (which uses up to 50 iterations, if I am not mistaken). Five iterations is insufficient for LDPC codes. For instance, with the CCSDS (128,64) code, it is reasonable to hypothesize that increasing from 5 to 200 iterations would yield performance exceeding 13.43 dB at 5 dB SNR-likely surpassing the reported SB-ECC results (though I cannot provide a direct citation to a BER based publication). Published results from other work also show significant improvements: 15 iterations achieve [7.32, 10.83, 15.43] dB (Choukroun et al. https://openreview.net/forum?id=XCP0MOMLPo), substantially better than the 5-iteration baseline. For fair comparison, either report BP decoder performance with a higher iteration count (e.g., 50-200 iterations) that better reflects its capabilities, Or provide a comprehensive complexity analysis demonstrating that SB-ECC has computational cost comparable to 5 BP iterations. While the performance gap with BCH and Polar codes is understandable (these codes are not optimized for BP), the comparison with LDPC codes requires particular caution since BP decoding is theoretically optimal for LDPC. The current setup may inadvertently handicap the BP baseline. The comparison with other model-free approaches (cross-MPT and DDECC) appears fair, as these comparisons are conducted under similar conditions, unlike the BP comparison which overlooks important factors such as iteration count, complexity, and code family characteristics (although it’s fair to recognize other paper did the same comparison).
- No computational complexity analysis is provided. Based on the transformer architecture, SB-ECC likely requires much more decoding operations than BP.
- The paper lacks sufficient ablation studies to identify the source of performance gains of SB-ECC over DDECC. It remains unclear whether the improvements stem from the different diffusion formulation, the modified input representation, the architectural changes, or other factors. Additional ablations isolating these components would strengthen the paper and provide valuable insights into which design choices contribute most to the observed performance improvements.
- The paper also reports performance using the negative natural logarithm of the BER which is unconventional for a coding paper and unnecessarily complicates unnecessarily the interpretation of the results. Standard practice in coding literature involves presenting BER or ideally block error rate (BLER) results directly.
- BLER would allow comparison with other published baselines, notably CCSDS (128,64) BP (200 Iterations) from Liva et al. (https://arxiv.org/pdf/1610.00873) which possibly surpass the performance of the SB-ECC considering a larger number of BP iterations.

**Presentation:** The paper is well-written with clear structure and comprehensive literature review. The mathematical exposition is good and the narrative flows well. I note the following weaknesses:
- The backbone architecture relies heavily on external references while lacking sufficient detail within the paper for straightforward reproduction. Although the authors commendably provide access to the code in the supplementary material, a more detailed description would strengthen the paper. It would be beneficial to explicitly emphasize in the CrossMPT table that this serves as the backbone model for the SB-ECC approach.
- The paper does not sufficiently detail the differences between SB-ECC and DDECC (Choukroun & Wolf 2022a). The distinction between the different diffusion formulations remains unclear, as do the practical implications of choosing one approach over the other. Additionally, potential backbone differences should be explicitly discussed.
- Experimental setup partially stated (number of BP iterations should be clearly stated)

Despite these issues, the paper is well written, and main ideas are communicated effectively.

**Significance:** The problem is important with broad practical relevance (ECCs are fundamental to communication systems). However, actual significance of the work is limited by the incremental contribution building on DDECC (2022a) and CrossMPT (2024), as well as the unclear practical advantage without complexity analysis and/or updated BP baseline.

**Originality:** The work combines existing ideas competently but incrementally. The idea of diffusion for decoding (DDECC) and the used backbone architecture (CrossMPT) are known, but their combination is novel making it an original contribution.

---

> ### Author Rebuttal · Authors · 2026-03-31
>
> We thank the reviewer for the very detailed and thorough review.
>
> **Weakness 1 + Question 1 - BP Fairness and Stronger Classical Baselines**
>
> Thank you for this important point, We added a deeper BP evaluation for the LDPC codes using stronger BP baselines with higher iteration counts (50,100,200), the results reported in the submission are all with 50 iteration from CrossMPT paper. In our simulation of CCSDS(128,64) the BLER align with the graph in (Liva et al. https://arxiv.org/pdf/1610.00873). The BP configuration should also have been stated explicitly in the paper, and we will correct this in the revised version.
>
> BER/BLER vs. $E_b/N_0$ plots are the standard and most interpretable presentation in coding theory, and the revised manuscript will reflect this more clearly. A related concern was also raised by Reviewer "Edfx". In response, we generated BER and BLER curves and will incorporate them into the revised manuscript so that the main comparisons are presented in the standard coding-theory format.
> We also provide BER and BLER graphs for the LDPC codes as requested:
>
> * LDPC$(49,24)$: BER: https://ibb.co/Jw5zxjVh, BLER: https://ibb.co/hxVmg72Q
> * LDPC$(121,70)$: BER: https://ibb.co/hRP0LNWx, BLER: https://ibb.co/zh9hP4Yg
> * CCSDS$(128,64)$: BER: https://ibb.co/TBKDM3nh, BLER: https://ibb.co/GvMcx4bw
>
> **Complexity and Practical Comparison to BP**
>
> We also agree that complexity is the complementary part of a fair comparison to stronger BP baselines. On the training side, SB-ECC has essentially the same training complexity as CrossMPT, since it uses the same backbone and still performs a single network evaluation per training sample. Relative to DDECC with the same backbone, the training setup is also very similar, with the main difference being the time-embedding overhead in the DDECC method.
>
> At inference time, we believe the most informative comparison is the controlled one that keeps the backbone fixed, since our goal is to isolate the effect of the proposed score-based formulation rather than conflate it with architectural differences. We will therefore add direct latency and throughput comparisons against CrossMPT and DDECC with the same CrossMPT backbone in the revised manuscript. We will also make clearer how the stopping-iteration statistics relate to effective inference cost in practice: each solver step corresponds to one network evaluation, so the stopping profile provides a useful proxy for decoding cost.
>
> **Question 2 + Weakness 3 - Ablations Isolating Gains Over DDECC**
>
> This concern was also raised by Reviewer "qerk". To address it, we added a controlled comparison that keeps the CrossMPT backbone fixed and compares DDECC against SB-ECC directly. In addition, the signed-vs.-magnitude ablation already included in the submission isolates the effect of the input representation while keeping the score-based framework fixed.
>
> In the revised manuscript, we will make this decomposition more explicit in both the main discussion and the ablation section.
>
> **Question 3 - Relation to Joint Encoder-Decoder Learning**
>
> Those works study a different setting, where the code and decoder are optimized together, whereas our paper focuses on improving the decoder while keeping the code fixed.
> At the same time, we believe the two directions are complementary rather than competing. If a stronger learned decoder is beneficial in a fixed-code setting, it may also be beneficial when combined with jointly learned code-decoder pipelines like the usage of hyper-networks (https://arxiv.org/pdf/2103.11780) or as the decoder block in a foundation model (https://openreview.net/pdf?id=7KDuQPrAF3) . More broadly, one conceptual takeaway of our work is that the standard $|y|$ preprocessing assumption might not be optimal in all neural decoding settings, where preserving the signed geometry of the observation can improve performance.
>
> **Backbone Description and Distinction from DDECC**
>
> The revised manuscript will make the role of the CrossMPT backbone more explicit, including which components are inherited directly and which are specific to SB-ECC.
>
> We will also clarify the distinction between SB-ECC and DDECC more directly. DDECC follows the standard reliability-based preprocessing and operates on $[|y|, s(y)]$, while SB-ECC learns directly from the signed observation $y$. In addition, DDECC uses a discrete reverse-diffusion process with syndrome-based line search, whereas SB-ECC uses a time-unconditional continuous score-based / PF-ODE formulation with solver-based inference and early stopping.
>
> **Experimental Details**
>
> The experimental setup will be stated more explicitly in the revised manuscript. In particular, the BP iteration counts will be given directly in the paper, and the experimental-details section will be revised to make the setup clearer and more complete.

---

> > ### Author Rebuttal · Reviewer_LjXg · 2026-04-01
> >
> > We thank the authors for the comprehensive and detailed rebuttal addressing all the concerns raised in our review. We acknowledge and appreciate the clarifications and additional results. In particular:
> > - The new results with higher BP iteration counts for better baseline fairness, the commitment to using conventional coding theory notation, and the clarification of the performance w.r.t. CCSDS (128,64) under higher iteration counts effectively address our concern about potential baseline underestimation.
> > - The improved complexity analysis through latency measurements. While a direct comparison with BP complexity would be interesting, we recognize this is not the focus of this work (model-free approach) and is partially addressed by the performance plots with increased BP iterations.
> > - The improved ablation studies regarding the origin of the observed performance gains.
> > - The commitment to clarify the role of the backbone architecture, the distinction between SB-ECC and DDECC, and the experimental setup in general, which will significantly improve reproducibility.
> >
> > As a general note, we also found the other rebuttals informative and interesting, and appreciate the inclusion of additional results for longer codes, which further strengthen the evaluation of scalability. We appreciate the authors' commitment to incorporate these improvements in the revised manuscript.

---

> > > ### Author Response · Authors · 2026-04-06
> > >
> > > Thank you for the thoughtful acknowledgement  and for the positive assessment. We appreciate your careful reading of the rebuttal and your recognition of the additional analyses and clarifications.

---

### Official Review · Reviewer_Edfx · 2026-03-08

**Soundness:** 3
**Presentation:** 3
**Significance:** 2
**Originality:** 3
**Overall Recommendation:** 5
**Confidence:** 5

**Summary:**

This paper advances the error correction code transformer (ECCT) literature, specifically building on diffusion denoising error correction codes (DD-ECC). It fundamentally changes the modeling approach by replacing the traditional magnitude-only representation of the received vector, \( |y| \), with the continuous, full received vector, \( y \). Additionally, the proposed method uses a score estimation-based approach instead of diffusion denoising and also eliminates the need for explicit SNR information during decoding by employing a time-unconditional score model trained across a range of noise levels with a linear noise schedule.

**Compliance With Llm Reviewing Policy:**

Affirmed.

**Ethical Review Concerns:**

Please refer to the weaknesses

**Final Justification:**

Within the field of model-free ECC neural decoders, this paper presents a novel approach to represent the input in a more meaningful format than magnitude and syndrome, which has been the de facto standard in recent years. Apart from that, the empirical results are thorough and convincing, resulting in an overall meaningful contribution.

**Key Questions For Authors:**

Please refer to the weaknesses

**Limitations:**

LImitations were not discussed

**Strengths And Weaknesses:**

### Strengths
- Changing the modeling choice from a magnitude and syndrome-based representation to a continuous real-valued representation is an interesting and refreshing approach. After seeing a long line of papers focused solely on improving the architecture and efficiency of ECCT models, this is a very welcome shift.
- The application of score-based modeling for error correction code transformers is a novel and compelling direction for the field.
- The authors provide a strong set of evaluations and ablations that effectively demonstrate the merit of the proposed components.

### Weaknesses
- Presenting results as a log BER table is counter to standard practice in information and coding theory, where the expectation is always to see BER and BLER vs. SNR plots on a logarithmic y-scale. This reliance on tables is a well-known trend in the ECCT line of work that needs to be corrected permanently. Instead of placing a couple of BER plots in the appendix, the main results should feature standard BER/BLER plots.
- There is a similar issue regarding the lack of comparison against code-specific classical decoders (such as Successive Cancellation List for Polar codes). Although it is understood that model-free neural decoders still lag classical decoders in certain cases, it is crucial to establish exactly where the field stands. Industry-standard classical decoder results should always be included in the baseline plots.
- A direct complexity comparison, covering both training and inference, against existing model-free neural decoders as well as classical non-neural decoders, is missing. I feel it's essential to compare the training time and convergence speed, etc., for score-based models vs diffusion denoising models.
- It would be highly valuable to see an explicit ablation isolating the representation change from the modeling change. Specifically, an ablation applying score estimation to the traditional \( (|y|, s(y)) \) representation, rather than the continuous \( y \), would help clarify the exact performance delta between the two representations.
- "Training details" do not specify the exact config used for training each code (ex. number of epochs trained). I parsed through the provided codebase and did not find any, except for an example training script. It would be good to include the exact config files for each coding scheme presented in the paper, and a single script that reproduces the results exactly.

### Minor Issues
- The acronym PF-ODE is first used on page 2 but is not defined until page 5.
- The acronym DPM is used but is never defined in the text.
- In general, while the results are very interesting from a coding theoretic point of view, specific contributions that a broad set of people working in AI might find interesting are missing.

---

> ### Author Rebuttal · Authors · 2026-03-31
>
> We thank the reviewer for the careful reading and constructive feedback.
>
> **Weakness 1 - Standard BER/BLER Presentation**
>
> The tables were originally intended to provide a compact summary across many code families and operating points, but we agree that the main experimental presentation should follow the standard coding-theory format of BER/BLER vs. $E_b/N_0$ curves. To address this, we generated BER and BLER curves and will include them in the main paper.
> Some samples can be found here:
> 1. LDPC(49, 24): BER - https://ibb.co/Jw5zxjVh, BLER - https://ibb.co/hxVmg72Q
> 2. LDPC(121, 70): BER - https://ibb.co/hRP0LNWx, BLER - https://ibb.co/zh9hP4Yg
>
> **Weakness 2 - Comparison to Code-Specific Classical Decoders**
>
> While our main goal is to compare against strong model-free neural decoders, code-specific classical decoders are essential reference baselines for understanding where these learned methods stand more broadly.
>
> For reference, we also include a comparison to Polar-specific classical decoders using SCL with sizes $L = 1$ and $L = 4$. This provides useful context, since SCL is a highly optimized decoder for Polar codes, whereas SB-ECC is a model-free neural decoder designed to operate within a broader learned-decoding framework. As expected, the SCL baselines are very strong on Polar codes. The plot shows that SB-ECC remains competitive across the evaluated Polar settings, and in some cases is close to or matches the stronger SCL baseline.
>
> Figures can be found at: https://ibb.co/4wQnP6J7
>
> **Weakness 3 - Training and Inference Complexity**
>
> A meaningful complexity comparison should separate training cost from inference cost, since the practical differences are not the same in the two stages.
>
> On the training side, CrossMPT, SB-ECC, and DDECC are very similar computationally, since all use a single network evaluation per training sample and the same underlying architecture. To make this comparison explicit, we summarize the average training epoch time for SB-ECC, DDECC, and CrossMPT on several representative codes.
>
> | Code           | CrossMPT |   DDECC |  SB-ECC |
> | -------------- | -------: | ------: | ------: |
> | BCH$(63,36)$   |  35.33 s | 34.89 s | 35.12 s |
> | POLAR$(64,32)$ |  35.94 s | 36.10 s | 35.86 s |
> | LDPC$(121,60)$ |  57.80 s | 58.02 s | 57.99 s |
>
> This makes the point clearly: the average training epoch times of CrossMPT, DDECC, and SB-ECC are nearly identical across different codes and different lengths.
>
> On the inference side, the distinction is more substantial. CrossMPT performs one forward pass, while SB-ECC and DDECC are iterative decoding methods. For this reason, the most informative inference comparison is the controlled same-backbone. We therefore report direct latency and throughput(samples/sec) comparisons against CrossMPT and DDECC under identical hardware settings. The example below illustrates this trade-off on BCH$(63,45)$. For clarity, the DDECC results are measured using the public repository implementation without early stopping.
>
> | Method   | Latency 3 dB (ms) | Latency 4 dB (ms) | Latency 5 dB (ms) | Throughput 3 dB | Throughput 4 dB | Throughput 5 dB |
> | -------- | ----------------: | ----------------: | ----------------: | --------------------------: | --------------------------: | --------------------------: |
> | SB-ECC   |49.96 |30.76 |18.34 |20.02 |32.51 |54.53 |
> | CrossMPT |14.69 |14.55 |14.48 |68.08 |68.71 |69.04 |
> | DDECC    |266.06 |266.85 |265.10 |3.76 |3.75 |3.77 |
>
> A more comprehensive latency/throughput comparison across additional codes and operating points will be included in the revised manuscript.
>
> **Weakness 4 - Ablation Isolating Representation from Modeling**
>
> This ablation is already included in the paper in Table 4. There, the score-based framework is kept fixed, and only the input representation is changed from the signed observation $\mathbf{y}$ to the magnitude-based input $|\mathbf{y}|$. We will make this more explicit in the revised manuscript.
>
> **Weakness 5 - Training Details and Reproducibility**
>
> Thank you for pointing this out. We will make the training setup more explicit in the revised manuscript, including the relevant per-code configuration choices. We will also release the full training configurations together with the codebase, so that the reported results can be reproduced more directly. The train.sh file in the supplementary material should work with the small change of batch_size=128
>
> **Minor Presentation Fixes**
>
> Thank you for catching these presentation issues. We will define PF-ODE and DPM at first use in the revised manuscript.
>
> **Relevance to the Broader AI Audience**
>
> Beyond the coding-theoretic results, the paper shows how score-based continuous denoising can be applied to a highly structured discrete inference problem, and highlights that preserving the signed geometry of the observation can be important when learning a continuous denoising field. We will make the broader AI relevance more explicit in the revised manuscript.

---

> > ### Author Rebuttal · Reviewer_Edfx · 2026-04-03
> >
> > I believe the author's rebuttal satisfactorily addressed my questions; I am raising the score accordingly. Specifically, I would like to emphasize that the authors should try to explicitly highlight the general applications of the proposed method, to appeal to the broader ICML community.

---

> > > ### Author Response · Authors · 2026-04-06
> > >
> > > Thank you for the thoughtful follow-up and for raising your score. We appreciate the positive assessment and the helpful suggestion.

---

### Official Review · Reviewer_qerk · 2026-03-13

**Soundness:** 3
**Presentation:** 3
**Significance:** 2
**Originality:** 2
**Overall Recommendation:** 4
**Confidence:** 3

**Summary:**

This paper formulates transformer-based error-correcting code (ECC) decoding as a continuous-time denoising process using a score-based generative model. The decoder integrates a probability-flow ordinary differential equation (PF-ODE) to iteratively denoise the received channel observation toward a valid codeword. Unlike many prior neural decoders that rely on magnitude-based preprocessing, the proposed approach directly uses the signed received vector as input and learns to predict continuous additive noise rather than log-likelihood ratios (LLRs). During inference, the number of denoising steps can be adjusted, providing a flexible latency–accuracy trade-off.

**Compliance With Llm Reviewing Policy:**

Affirmed.

**Final Justification:**

The authors provided the evalutation results supporting that the proposed training method is effective for ECCT and CrossMPT. These results resolved my concerns, so I have changed my recommendation from [weak reject] to [weak accept].

**Key Questions For Authors:**

- Prior neural decoders often use magnitude-based preprocessing (e.g., $|y|$ together with syndrome information) to reduce overfitting and exploit channel symmetry. In contrast, this work directly uses the signed received vector as input. Could the authors clarify the reasons how to prevent overfitting in this setting?

- The following work has explored consistency flow models for ECC decoding, which aim to achieve one-step denoising without iterative diffusion sampling. Since the proposed method relies on multi-step score-based denoising, a comparison with this approach would help clarify the advantages of the proposed framework.

H. Lei, “Consistency Flow Model Achieves One-step Denoising Error Correction Codes,” https://openreview.net/forum?id=ZyLQVFDgfr

**Limitations:**

yes

**Strengths And Weaknesses:**

## Strengths

- This paper introduces a score-based diffusion framework for ECC decoding, casting decoding as continuous denoising guided by parity constraints.
- The paper highlights the difficulty of learning continuous denoising directions when magnitude-based preprocessing is used, and instead proposes using the signed channel observation directly as input.
- The use of early stopping in the PF-ODE integration allows the decoder to reduce latency while maintaining competitive decoding performance.

## Weaknesses

- The main contribution of the paper lies in the proposed **training and inference framework** based on score-based generative modeling rather than in the neural network architecture itself. However, the experimental comparison in Table 1 mixes different architectures and training paradigms, making it difficult to isolate the effect of the proposed method. Currently, Table 1 compares:
    - CrossMPT: CrossMPT architecture + standard training
    - DDECC: ECCT architecture + denoising diffusion training
    - SB-ECC: CrossMPT architecture + proposed score-based training

    Since the paper primarily proposes a new **training and inference methodology**, a more controlled comparison would be desirable. For example:

    - DDECC vs. ECCT + proposed training
    - CrossMPT + denoising diffusion training vs. CrossMPT + proposed training

    Such comparisons would more clearly demonstrate whether the performance gains stem from the proposed training framework rather than architectural differences.

- The experimental evaluation focuses primarily on relatively short block lengths ($n \le 128$). It would be helpful to see results on longer codes to better understand the scalability of the proposed approach, since neural decoders often face increasing difficulty as the block length grows.

---

> ### Author Rebuttal · Authors · 2026-03-31
>
> We thank the reviewer for the careful reading and constructive feedback.
>
> **Weakness 1 - Controlled Comparisons to Isolate Methodology from Backbone**
>
> Since the contribution of the paper is the score-based decoding framework rather than a new neural architecture, the evaluation should distinguish methodological gains from backbone effects. To address this, we added a same-backbone comparison that fixes the CrossMPT architecture and compares DDECC training directly against SB-ECC. This isolates the contribution of the proposed score-based training and PF-ODE decoding from architectural differences.
>
> | Code Type | Code Params | DDECC (4) | DDECC (5) | DDECC (6) | SB-ECC (4) | SB-ECC (5) | SB-ECC (6) |
> | --------- | ----------: | --------: | --------: | --------: | ---------: | ---------: | ---------: |
> | BCH|(63,36) |5.26 |7.24 |10.05 |   **5.74** |   **8.12** |  **11.20** |
> | BCH|(63,45) |5.67 |8.07 |11.41 |**6.58** |   **9.48** |  **13.17** |
> | BCH|(63,51) |5.37 |7.48 |10.51 |**6.19** |   **8.82** |  **12.45** |
> | POLAR|(64,32) |7.03 |9.69 |12.97 |**7.77** |  **10.30** |  **13.78** |
> | POLAR|(64,48) |6.07 |8.40 |10.90 | **6.63** |   **8.64** |  **11.27** |
> | POLAR|(128,64) |8.16 |12.04 |-- |**9.03** |  **13.13** |16.94 |
> | POLAR|(128,86) |7.92 |11.34 |15.61 |**8.03** |  **11.48** |  **16.10** |
> | POLAR|(128,96) |7.24 | **10.32** |13.26 |   **7.64** |  **10.32** |  **13.37** |
> | LDPC|(49,24) |6.26 |9.07 |12.71 |   **7.74** |  **10.88** |  **14.63** |
> | LDPC|(121,60) |6.22 |10.18 |-- |   **6.33** |  **10.38** |16.38 |
> | LDPC|(121,70) |7.64 |12.30 |-- |   **7.75** |  **12.69** |19.24 |
> | LDPC|(121,80) |8.62 |13.55 |-- |   **8.84** |  **14.02** |20.42 |
> | MacKay    |(96,48) |8.58 |12.48 |-- |   **8.97** |  **12.82** |16.25 |
> | CCSDS     |(128,64) |8.40 | 13.18 |-- |**8.60** |**13.43** |19.20 |
>
> Empty cells (`--`) indicate results that are still being finalized and will be updated when available. In this controlled setting, SB-ECC still outperforms the diffusion-based alternative, supporting the claim that the observed gains are not due only to the backbone, but also to the proposed formulation and inference procedure.
>
> We will also add to the revised manuscript DDECC vs SB-ECC with the original ECCT backbone. Together, these controlled experiments provide a much cleaner test of whether the performance gains stem from the proposed methodology rather than from architecture choice.
>
> **Weakness 2 - Scalability to Longer Block Lengths**
>
> To address this, we added experiments on longer codes, including LDPC$(204,102)$ and LDPC$(529,440)$. The BP results here are with 50 iterations. More codes (e.g., POLAR$(512,384)$) and models (e.g., DDECC with CrossMPT backbone) are currently being trained, and we will update with results when available.
>
> **Longer-block evaluation.** We report $-\ln(\mathrm{BER})$ at $E_b/N_0 \in {4,5,6}$ dB.
>
> | Code Type | Code Params | BP(4) | BP(5) | BP(6) | SB-ECC(4) | SB-ECC(5) | SB-ECC(6) |
> | --------- | ----------: | ----: | ----: | ----: | --------: | --------: | --------: |
> | LDPC  | $(204,102)$ | 10.27 | 13.79 | 16.29 | 11.20 |15.78 |18.77 |
> |            | $(529,440)$ |  8.18 | 14.88 |-- |7.88 |15.21 |-- |
>
> **Question 1 - Why Signed Observations Do Not Lead to Overfitting**
>
> In prior neural decoders, magnitude-based preprocessing is often used to exploit channel symmetry and reduce overfitting. In our setting, however, the model is not predicting bit logits or LLRs, but learning a continuous denoising vector field in $\mathbb{R}^n$. For this task, the sign of each coordinate in the received vector carries essential directional information, and mapping $\mathbf{y} \mapsto |\mathbf{y}|$ removes information that is directly relevant to the denoising direction.
>
> In practice, overfitting is mitigated by several factors. First, the model is trained to predict additive Gaussian noise, which is a smoother regression target than directly predicting codewords or hard decisions. Second, the backbone retains strong structural inductive bias through Tanner-graph masking and parity-based conditioning. Finally, our signed-vs.-magnitude ablation shows that replacing $\mathbf{y}$ with $|\mathbf{y}|$ while keeping the rest of the pipeline fixed causes a large performance drop, indicating that the signed observation is a necessary part of learning the denoising field.
>
> **Question 2 - Relation to One-Step Consistency-Flow Decoding**
>
> Thank you for pointing us to this relevant work. We will add it to the revised manuscript and position it more explicitly relative to SB-ECC. The two approaches target different operating regimes: consistency-flow methods aim at one-step decoding, whereas SB-ECC focuses on a tunable accuracy-latency trade-off through multi-step score-based decoding with early stopping.
> In the following link you can find the results comparison between the methods - https://ibb.co/TDGywFdX

---

> > ### Author Rebuttal · Reviewer_qerk · 2026-04-02
> >
> > The rebuttal partially addresses the concern by providing some additional results (e.g., DDECC vs. ECCT with the proposed training). However, key issues remain insufficiently resolved for the more advanced CrossMPT backbone. In particular, a controlled comparison between CrossMPT with denoising diffusion training and CrossMPT with the proposed score-based training is important to clearly distinguish methodological gains from backbone effects. The original manuscript did not sufficiently isolate these factors, and this limitation remains. Similarly, for longer block-length experiments, it would be important to provide comparisons under the same controlled setting.

---

> > > ### Author Response · Authors · 2026-04-05
> > >
> > > **Controlled Same-Backbone CrossMPT Comparison**
> > >
> > > Thank you for the follow-up and for taking the time to engage with our rebuttal in detail. We appreciate the clarification, and we would like to restate this point more explicitly.
> > >
> > > To isolate methodology from backbone effects, we performed the controlled same-backbone comparison requested by the reviewer: the table below compares our method, SB-ECC with the **CrossMPT backbone**, against DDECC with the **same CrossMPT backbone**. To obtain the DDECC results in this controlled setting, we used the DDECC repository together with the CrossMPT backbone implementation, and trained the model according to the training setting reported in DDECC. This comparison is intended to isolate the effect of the decoding/training formulation while keeping the backbone fixed.
> > >
> > > We have now added more results as additional runs finished:
> > >
> > > |  |  | DDECC (CrossMPT) | | | SB-ECC (CrossMPT) |  |  |
> > > |---|---|---:|---:|---:|---:|---:|---:|
> > > | Code Type | Code Params | 4 dB | 5 dB | 6 dB | 4 dB | 5 dB | 6 dB |
> > > | BCH | (63,36) | 5.30 | 7.32 | 10.25 | 5.74 | 8.12 | 11.20 |
> > > | BCH | (63,45) | 5.67 | 8.07 | 11.41 | 6.58 | 9.48 | 13.17 |
> > > | BCH | (63,51) | 5.37 | 7.48 | 10.51 | 6.19 | 8.82 | 12.45 |
> > > | POLAR | (64,32) | 7.03 | 9.69 | 12.97 | 7.77 | 10.30 | 13.78 |
> > > | POLAR | (64,48) | 6.07 | 8.40 | 10.90 | 6.63 | 8.64 | 11.27 |
> > > | POLAR | (128,64) | 8.16 | 12.04 | 16.27| 9.03 | 13.13 | 16.94 |
> > > | POLAR | (128,86) | 7.92 | 11.34 | 15.61 | 8.03 | 11.48 | 16.10 |
> > > | POLAR | (128,96) | 7.24 | 10.32 | 13.26 | 7.64 | 10.32 | 13.37 |
> > > | LDPC | (49,24) | 6.26 | 9.07 | 12.71 | 7.74 | 10.88 | 14.63 |
> > > | LDPC | (121,60) | 6.22 | 10.18 | 15.89| 6.33 | 10.38 | 16.38 |
> > > | LDPC | (121,70) | 7.64 | 12.30 | 17.98 | 7.75 | 12.69 | 19.24 |
> > > | LDPC | (121,80) | 8.62 | 13.55 | 18.26 | 8.84 | 14.02 | 20.42 |
> > > | MacKay | (96,48) | 8.58 | 12.48 | 16.04 | 8.97 | 12.82 | 16.25 |
> > > | CCSDS | (128,64) | 8.40 | 13.18 | 17.47 | 8.60 | 13.43 | 19.20 |
> > >
> > > For convenience, if the table does not render clearly in OpenReview, we also provide an image version here:  https://ibb.co/d05nDr8r
> > >
> > > We also agree that the longer block-length evaluation should be presented under the same controlled setting, and we are extending the corresponding same-backbone comparisons to longer codes as well after your feedback. The longer-code experiments are currently running, since these models require substantially longer training and inference time, the additional results—including LDPC/Polar classical baselines and the controlled DDECC(CrossMPT backbone) vs. SB-ECC(CrossMPT backbone) comparison will be added.

---

### Decision · Program_Chairs · 2026-04-30

**Decision:**

Accept (regular)

**Comment:**

The reviewers agree that the score-based diffusion framework for decoding is a interesting idea, and note the novelty of using the raw signed channel observation as input for training. They have also identified a few important weaknesses in the orginal manuscript (mixing different architectures and training paradigms, comparison with relatively weak BP and classical baselines etc.) which have been effectively addressed in the rebuttal phase. The additional discussion and experimental comparisons presented in the rebuttal should be added to the final paper, which will make it stronger and more comprehensive.